# A reconfigurable binary/ternary logic conversion-in-memory based on drain-aligned floating-gate heterojunction transistors

Chungryeol Lee[1], Changhyeon Lee[1], Seungmin Lee[1], Junhwan Choi [2], Hocheon Yoo [3,5] ✉ & Sung Gap Im [1,4,5] ✉

A new type of heterojunction non-volatile memory transistor (H-MTR) has been developed, in which the negative transconductance (NTC) characteristics can be controlled systematically by a drain-aligned floating gate. In the H-MTR, a reliable transition between N-shaped transfer curves with distinct NTC and monolithically current-increasing transfer curves without apparent NTC can be accomplished through programming operation. Based on the H-MTR, a binary/ternary reconfigurable logic inverter (R-inverter) has been successfully implemented, which showed an unprecedentedly high static noise margin of 85% for binary logic operation and 59% for ternary logic operation, as well as long-term stability and outstanding cycle endurance. Furthermore, a ternary/binary dynamic logic conversion-in-memory has been demonstrated using a serially-connected R-inverter chain. The ternary/binary dynamic logic conversion-in-memory could generate three different output logic sequences for the same input signal in three logic levels, which is a new logic computing method that has never been presented before.

As visual/speech recognition, smart healthcare systems, and other personalized artificial intelligence (AI) technologies become ubiquitous in daily life, the demand for compatibility in human-machine interfaces, as well as the information processing capability of integrated circuits (ICs), is increasing enormously[1–3]. Organic ICs based on organic thin-film transistors (OTFTs) are highly promising candidate for the realization of such intelligent Internet of Things (IoT) devices due to their light-weight, intrinsic flexibility and compatibility with various form factors, which can facilitate user-friendly edge interaction by integrating with wearable sensors and displays[4–6]. Over the past two decades, there have been numerous intensive efforts in material

engineering or device optimization to boost the performance of OTFTs[7–11]. Notwithstanding the successful advancements in various applications such as logic circuits, physical/chemical sensors, memory and artificial synapse, low resistance of organic materials to high temperatures and solvents makes OTFTs incompatible with conventional photo-lithography process, limiting the integration density and information processing capability of organic ICs[12].

Reconfigurable electronics can be a promising breakthrough for such scaling issues[13]. Through the use of dynamically modifiable logic functions during circuit operation, the reconfigurable logic can afford more diverse and complex calculations within a given footprint. The

[1]Department of Chemical and Biomolecular Engineering, Korea Advanced Institute of Science and Technology (KAIST), 291 Daehak-ro, Yuseong-gu 34141, Korea. [2]Department of Chemical Engineering, Dankook University, 152, Jukjeon-ro, Suji-gu, Yongin 16890, South Korea. [3]Department of Electronic Engineering, Gachon University, 1342 Seongnam-daero, Seongnam 13120, Korea. [4]KAIST Institute for NanoCentury (KINC), Korea Advanced Institute of Science and Technology (KAIST), 291 Daehak-ro, Yuseong-gu 34141, Korea. [5]These authors contributed equally: Hocheon Yoo, Sung Gap Im. ✉e-mail: hyoo@gachon.ac.kr; sgim@kaist.ac.kr

key strategy of this circuit operation is to diversify the field-effect characteristics, thus realizing multi-functionality within a single transistor unit, which allows for reconfigurable logic to be implemented within a minimum circuit unit without involving complicated device architecture or hardware cluster configuration. Several reconfigurable logic devices have been recently implemented using various device architectures and state-of-the-art material systems, including Si[14,15], Ge[16,17], transition metal dichalcogenides (TMDs)[18–22], whose polarity can be dynamically toggled either to p-type or n-type. One desirable method for implementing such reconfigurable transistor is to integrate a memory function into the inherent switching function of the transistor by employing a charge storage layer. In this approach, the non-volatile state of the memory can dynamically control the amount of charge carriers (holes for p-type and electrons for n-type) injected into the channel, thereby modifying the polarity of the transistor[20]. Since memory-based reconfigurable logic circuits have both computation and data storage capabilities, they can also be utilized as logic-in-memory, which can address the von-Neumann bottleneck by reducing latency and energy burdens associated with data transmission between processing and memory units[23].

Along with the reconfigurable logic, multi-valued logic (MVL) has emerged as a promising approach for data-intensive applications[16,24]. Compared to conventional digital systems, the MVL systems use more than three logic states, enabling higher data processing efficiency with enhanced integration density within the same design rule. For example, ternary logic reduces the system complexity by ~63.1% compared to conventional binary logic systems[25]. MVL has been demonstrated by using heterojunction transistors (H-TRs) with various channel materials such as organic semiconductors[26–32], graphene[33–35], metal-oxides[36,37], and 2D transition metal dichalcogenides (TMDs)[38–50]. In H-TRs, two channel materials generate a point where the magnitude of electric current reaches its peak and then decreases at a specific gate bias range. This unique characteristic is termed as negative transconductance (NTC), which enables the construction of ternary logic systems by using conventional CMOS design while maintaining the required number of transistors to implement each logic state. Besides the MVL, it has been reported that many other devices such as frequency doubler, binary frequency shift keying and binary phase shift keying can also be implemented by harnessing the NTC characteristics of the H-TRs[51]. However, it has been quite challenging to adjust the NTC characteristics in a suitable form to fully exploit the advantages of H-TRs for the target applications. This is because the NTC characteristics mainly depend on the charge carrier density of two channels in H-TRs, which is under the control of same gate bias. In order to ensure the desired NTC characteristics, therefore, the channel materials to form heterojunction in the H-TRs must be selected with the consideration of their electrical properties such as intrinsic carrier density, carrier mobility, and interface with gate dielectric, simultaneously. These issues become more problematic when constructing ternary logic circuits, since the magnitude of electric current as well as the range of gate bias for NTC (NTC region) of H-TR need to be manipulated precisely with respect to the counterpart transistor in the ternary logic inverter to provide optimal intermediate logic state. Moreover, the limited noise margin of each state in ternary logic systems (33.3%) compared to that in binary logic (50%) highlights the importance of optimizing NTC characteristics[52]. However, most of the previously reported studies have shown non-symmetric in-out voltage transfer characteristics (VTC) for a ternary logic inverter due to uncontrolled NTC characteristics, which cannot define noise margin for the intermediate logic state to implement the sequential integration of the inverters, and thus restricts the integration level of ternary logic circuit to unit inverter (Supplementary Table 1).

In this study, we propose a novel type of heterojunction non-volatile memory transistor (H-MTR), which enables dynamic control of the NTC characteristics by incorporating the function of non-volatile memory into H-TR. Unlike the conventional flash-memory, we use a drain-aligned floating gate (FG) in the proposed H-MTR, which partially overlaps with only a portion of the channel. The partially overlapped FG is conceived from the asymmetric device configuration of H-TR, where the p-type layer connects the source to drain electrode, while the n-type layer interacts only with the drain electrode through the interposed p-type layer. By manipulating the magnitude of the gate-to-drain electric field ($E_{GD}$), the drain-aligned FG under the n-type semiconductor can effectively adjust the amounts of electrons injected into the channel, and thus enables the systematic control of NTC characteristics with programming operation. By using dynamic NTC characteristics, we successfully achieved high-performance binary/ternary reconfigurable inverter (R-inverter), which features not only high static noise margin (SNM) corresponding to 85% for binary logic operation and 59% for ternary logic operation of the ideal value, but also stable retention property and excellent cycle durability. To the best of our knowledge, the SNM of ternary logic operation realized in this study is even higher compared to that of other ternary logic circuits reported previously, which were based on devices with other working principles (Supplementary Table 2). Moreover, a direct connection between binary and ternary logic systems can be realized without involving complex circuits, since the binary/ternary reconfiguration does not result in non-symmetric VTC or require a change in the supply voltage ($V_{DD}$). For example, the proposed R-inverter could be cascaded to implement high-level circuits with all individual units fully reconfigurable. As a proof of concept, binary/ternary logic conversion-in-memory is demonstrated using a two-stage R-inverter for the first time, which produces output signals in three different sequences with each of three logic levels, and therefore can perform all the functions of a standard ternary inverter (STI), a positive ternary inverter (PTI), and a negative ternary inverter (NTI) according to the corresponding memory states of the constituent R-inverter.

## Results

### Design of the H-MTR

A schematic illustration of the proposed H-MTR is shown in Fig. 1a. The H-MTR has a typical structure of bottom-gate, top-contact flash memory transistor, except that the FG is located only under the drain electrode area. This drain-aligned FG reflects the operating condition of the H-MTR, where a contact electrode on the p-type single layer serves as the source and a contact electrode on the p-type/n-type double layer serves as the drain, in which case holes are injected from the source to the p-type semiconductor while electrons are injected from drain to the n-type semiconductor[32]. Therefore, the drain-aligned FG can control the amount of electron injection from the drain to the channel according to its memory state, while let the amount of hole injection from the source to the channel not perturbed. For blocking dielectric layer (BDL) and tunneling dielectric layer (TDL), poly(2-cyanoethyl acrylate-co-diethylene glycol divinyl ether) [p(CEA-co-DEGDVE)] (named pCD) and poly(1,3,5-trivinyl-1,3,5-trimethyl cyclotrisiloxane) (pV3D3) were employed, respectively, both of which were deposited by a vapor-phase polymer deposition process, termed initiated chemical vapor deposition (iCVD)[53–55]. The thickness of the pCD BDL and the pV3D3 TDL was 70 and 15 nm respectively, yielding capacitance per unit area ($C_i$) of ≈62 nF cm⁻² for BDL and ≈123 nF cm⁻² for TDL (Supplementary Fig. 1). Such ultrathin dielectric layers allow for the low-power operation of the H-MTR[56]. Note that the pCD BDL is high-$k$ dielectric ($k > 5$)[57] while the pV3D3 TDL is low-$k$ dielectric ($k \sim 2.2$)[58], which is advantageous for non-volatile memory with high gate coupling ratio[59]. This enables a higher electric field ($E$) to be applied mostly across the TDL than the BDL during the programming operation, facilitating the Fowler-Nordheim (F-N)-like tunneling through the TDL[58] while minimizing the charge leakage through the BDL (Supplementary Fig. 1). Dinaphtho[2,3-b:2′,3′-f]thieno[3,2-b]thiophene (DNTT) and *N,N′*-Ditridecyl-3,4,9,10-perylenetetracarboxylic

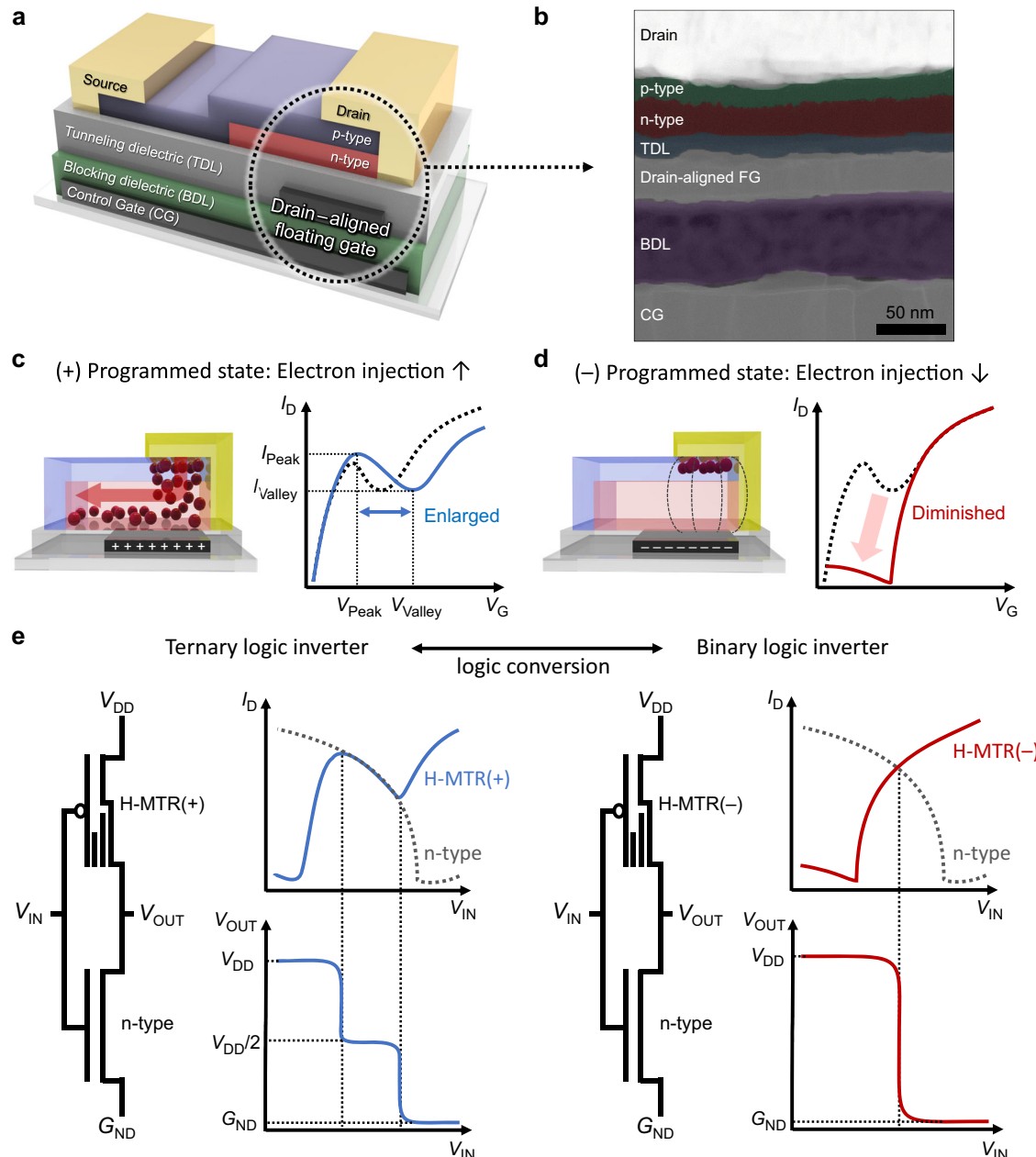

**Fig. 1 | Design of heterojunction non-volatile memory transistor (H-MTR) and binary/ternary reconfigurable logic inverter (R-inverter). a** A schematic illustration of the H-MTR. **b** A cross-sectional high-resolution transmission electron microscope (HRTEM) image of the H-MTR. False color modification was applied to distinguish each layer. **c** A schematic diagram illustrating the charge injection at drain electrode in the H-MTR with the (+) programmed state (left) and **d** (−) programmed state (right). Red sphere represents electron carrier. **e** A conceptual schematic illustrating the operating principle of the R-inverter.

diimide (PTCDI-C13) were used as p-type and n-type channel, respectively, which have been used widely for organic p-n heterojunction transistors[60,61]. The semiconducting layers with high charge mobility comparable to each other can readily form a heterointerface through a sequential vacuum evaporation process. The chemical structures of polymer dielectrics and organic semiconductors constituting the H-MTR, as well as corresponding energy band diagrams, are summarized in Supplementary Fig. 2. The vertical configuration along the FG side of the H-MTR was confirmed through the high-resolution transmission electron microscopy (HRTEM) (Fig. 1b). The cross-sectional HRTEM image showed the layers from control gate (CG) to the source/drain electrodes of the H-MTR, where the direct conducting paths from CG to FG and from FG to drain were blocked by BDL and TDL (Supplementary Fig. 3). Figure 1c shows the operating mechanism

of the H-MTR. If the FG becomes positively charged by applying the negative bias to CG (− programming operation, Fig. 1c), electron injection from the drain to the channel will be facilitated with the effectively increased $E_{GD}$. This enables a larger amount of electrons to be accumulated in the channel even at higher negative control gate voltage ($-V_G$), which in turn increases the magnitude of current and the range of NTC region. On the other hand, electron injection is limited by the effectively decreased $E_{GD}$ when the FG is negatively charged by applying the positive bias to CG (+ programming operation, Fig. 1d). This makes electron depletion for the whole range of $V_G$, and thus resulting in monolithically current-increasing transfer characteristics without NTC. Based on the capability to generate or eliminate the NTC characteristics of the H-MTR, the R-inverter can be implemented by connecting the H-MTR to n-type transistor (Fig. 1e, left). For example, if

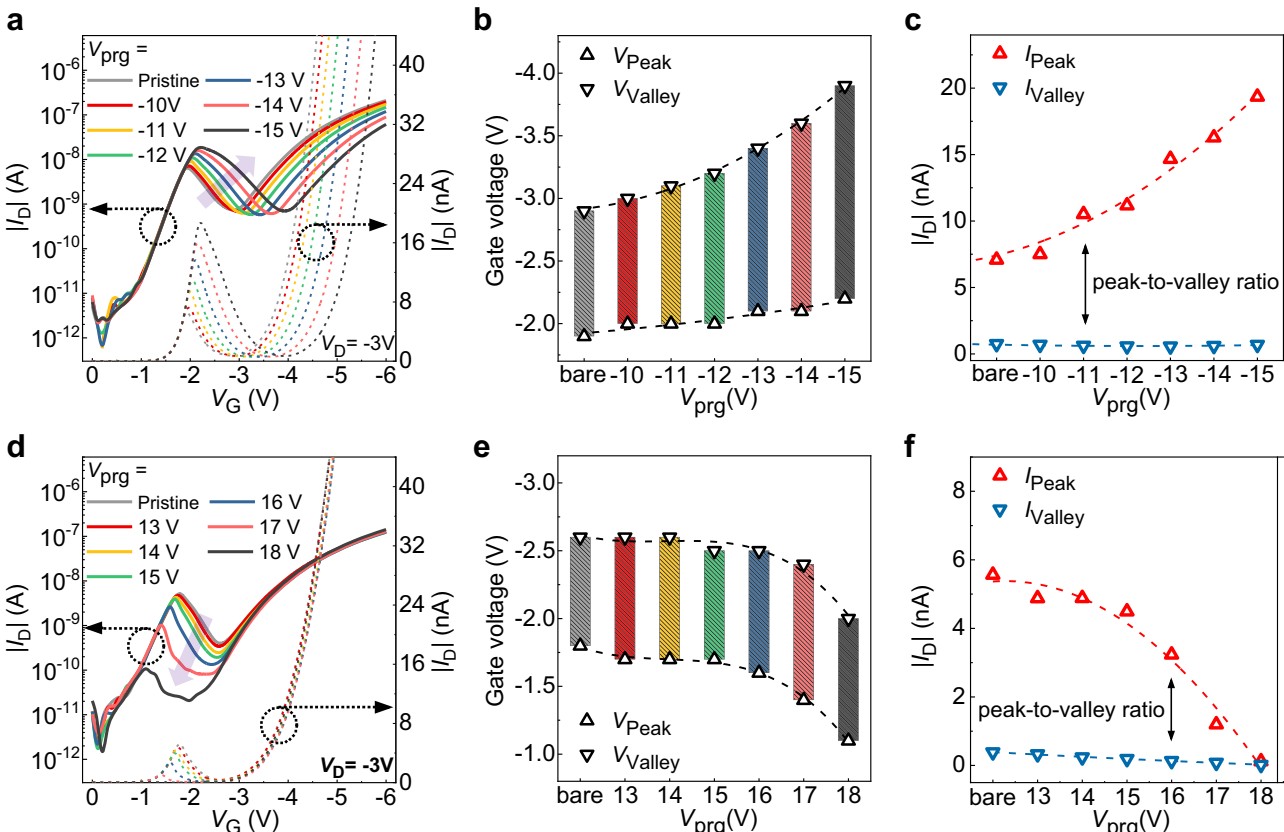

**Fig. 2 | Investigation of the device characteristics according to each programmed state. a** The transfer characteristics, **b** peak voltage ($V_{Peak}$) and valley voltage ($V_{valley}$), and **c** peak current ($I_{Peak}$) and valley current ($I_{Valley}$) of the H-MTR with respect to (−) programming voltage (−$V_{prg}$). **d** The transfer characteristics, **e** $V_{Peak}$ and $V_{valley}$, and **f** $I_{Peak}$ and $V_{Valley}$ of the H-MTR with respect to (+) programming voltage (+$V_{prg}$). Curves of different colors correspond to different $V_{prg}$.

the FG of H-MTR becomes positively charged, the H-MTR and n-type transistor can exhibit the comparable conductance in the NTC region, which makes a ternary logic inverter with the intermediate logic state. On the other hand, if the FG of H-MTR is negatively charged, the voltage at which the two transistors show the comparable conductance is limited to a few points, resulting in binary logic inverter without intermediate logic state (Fig. 1e, right).

**Dynamic NTC characteristics of the H-MTR**
The dynamic NTC characteristics of the H-MTR were investigated by applying negative (−$V_{prg}$) or positive (+$V_{prg}$) programming bias to the CG. For (−) programming operation, the electron injection from the drain to the channel was promoted and the range of NTC region enlarged gradually as the |$V_{prg}$| increased (Fig. 2a). For example, the range from the peak voltage ($V_{Peak}$), corresponding to $V_G$ with the highest drain current ($I_{Peak}$), to the valley voltage ($V_{Valley}$), corresponding to $V_G$ with the lowest drain current ($I_{Valley}$), in NTC region increased gradually from 1 V (pristine) to 1.7 V ($V_{prg}$ = −15 V) (Fig. 2b). Moreover, both $V_{Peak}$ and $V_{Valley}$ increased with higher |$V_{prg}$| as well; the $V_{Peak}$ increased from −2.9 V (pristine) to −3.9 V ($V_{prg}$ = −15 V) and $V_{Valley}$ increased from −1.9 V (pristine) to −2.2 V ($V_{prg}$ = −15 V). Here, the amount of $V_{Peak}$ shift is much larger than that of the $V_{Valley}$. This asymmetric shift of $V_{Peak}$ and $V_{Valley}$ with programming operation can be attributed to the device architecture of the H-MTR, where the top DNTT interfacing with PTCDI-C13 serves as a back channel, whose electrical characteristics would be affected by the charge carrier density in PTCDI-C13, and thereby would be quite different from the bottom DNTT in touch with dielectric layer, yielding two nominal threshold voltages ($V_{TH}$) in the transfer characteristics; The former one is originated from the charge carrier transport through the lateral

DNTT/PTCDI-C13 junction near the gate dielectric and corresponded to the $V_{TH}$ before the NTC region (namely, $V_{TH\_P1}$) while the latter stems from the charge carrier transport through the DNTT back channel, corresponding to the $V_{TH}$ exceeding the NTC region (namely, $V_{TH\_P2}$) (Supplementary Fig. 4). As described previously, the FG covered only the drain electrode area in the H-MTR, and there was no difference in the gate-to-source electric field ($E_{GS}$) before and after the programming operation. Accordingly, the amounts of holes injected from the source to the channel remained the same with no change in $V_{TH\_P1}$ throughout the applied $V_{prg}$. On the other hand, $E_{GD}$ could be screened (or promoted) with the positively (or negatively) charged FG, which then increased (or decreased) the amounts of electron injection from drain to channel depending upon the programming bias polarity. Hence, with −$V_{prg}$, the electron concentration in the channel would increase, resulting in $V_{TH\_P2}$ shift toward −$V_G$ direction while there was only negligible change in $V_{TH\_P1}$. Here, the $V_{TH\_P2}$ shift makes a parallel movement of V-shaped curve with $V_{Valley}$ as a center, since the stacked DNTT and PTCDI-C13 bilayer performs as an ambipolar semiconductor[62]. The extracted $I_{Peak}$ and $I_{Valley}$ from the transfer curves with respect to $V_{prg}$ also confirmed the parallel shift of V-shaped curve and support our explanation above (Fig. 2c). Compared to the $I_{Valley}$ which was practically constant with a value of about 0.6 nA, the $I_{Peak}$ gradually increased from 7 nA (pristine) to the 19 nA ($V_{prg}$ = −15 V) with higher $V_{prg}$, leading to the larger peak-to-valley current ratio. For (+) programming operation, electron injection from the drain was suppressed, and the NTC characteristics gradually diminished with the increasing $V_{prg}$ (Fig. 2d). The $V_{TH\_P1}$ remained at its initial value since there was no change in $E_{GS}$ before and after the programming operation, as mentioned previously. Note that only a marginal shift in $V_{TH\_P2}$ was observed as well because the FG was designed to avoid the lateral

p/n junction edge, which retained the energy barrier for hole transport through the DNTT back channel in the initial state. The fixed $V_{\text{TH\_P1}}$ and $V_{\text{TH\_P2}}$ with $+V_{\text{prg}}$ led to the gradual shift of the NTC region toward $+V_G$ direction without changing the size of the NTC range (Fig. 2e). For example, a value of $V_{\text{Peak}}$ changed from −1.8 V to −1.1 V and a value of $V_{\text{Valley}}$ changed from −2.6 V to −2 V with the $V_{\text{prg}}$ of 18 V, resulting in a quite similar NTC range (-0.9 V). Whereas, both $I_{\text{Peak}}$ and $I_{\text{Valley}}$ decreased by more than an order of magnitude (from nA to pA level) when the $V_{\text{prg}}$ surpassed 18 V (Fig. 2d, black line), which made the whole current level of the NTC region practically similar to that of the off-state (Fig. 2f). This allowed the H-MTR to maintain its off-state below the $V_{\text{TH\_P2}}$ and to operate as if it is a monolithically current-increasing transistor without the NTC, which can lead to binary logic operation without distinct intermediate logic state when integrated into an inverter. A schematic illustration and energy band diagram describing the charge carrier transport in H-MTR according to its memory state are provided in Supplementary Fig. 5.

The air stability and thermal stability of the H-MTRs were further investigated to examine their practical applicability for logic devices. The H-MTRs were encapsulated with 20 nm-thick $Al_2O_3$ layer via atomic layer deposition (ALD) process to implement more stable device in air ambient. The $Al_2O_3$-encapsulated H-MTRs were stored in air ambient (20 °C, 45% relative humidity) except during measurements. The initial electrical properties of the $Al_2O_3$-encapsulated H-MTRs were fully maintained during the entire measurement period of ~144 h (Supplementary Fig. 6). In terms of thermal stability, the NTC characteristics of the H-MTR were monitored by increasing the device temperature in increments of 20 K. Initially, both $I_{\text{Peak}}$ and $I_{\text{Valley}}$ increased gradually as the device temperature increased up to 358 K, due to thermally activated charge transport in organic semiconductors[63] (Supplementary Fig. 7). After 378 K, the $V_{\text{TH\_P1}}$ was gradually increased and resulted in decreased $I_{\text{Peak}}$, which is attributed to thermal degradation of the p-type semiconductor, DNTT[64]. At 478 K, the H-MTR no longer exhibited switching characteristics. Although the peak-to-valley current ratio was continuously decreased as the temperature increased, NTC characteristics could still be observed even for temperatures up to $T = 458$ K, which demonstrates the potential of the proposed device concept for logic device applications.

## Binary/ternary reconfigurable logic inverter

For the demonstration of the binary/ternary reconfigurable logic operation, the R-inverter was fabricated by connecting the H-MTR (pull-up transistor) to n-type transistor with PTCDI-C13 channel (pull-down transistor) (Fig. 3a, b). To examine the ternary logic operation, the VTC of the R-inverter was analyzed by applying $-V_{\text{prg}}$ to the H-MTR (Fig. 3c). At the initial state, the R-inverter showed three logic states ($V_{\text{DD}}$ for logic 2, -2.2 V for logic 1, and $G_{\text{ND}}$ for logic 0) owing to the distinct NTC region as well as full-swing output voltage ($V_{\text{OUT}}$) from $V_{\text{DD}}$ to $G_{\text{ND}}$ resulting from the high on/off current ratio ($I_{\text{on/off}}$) of the H-MTR[32]. Considering the operating voltage (6 V), however, $V_{\text{OUT}}$ of -2.2 V for the intermediate logic state was far below the optimal value ($V_{\text{DD}}/2 = 3$ V) and the range of 1.2 V for the intermediate logic state was also quite narrow compared to the range of other two logic states (2.6 V for logic 0 and 2.2 V for logic 2, respectively), which is not desirable for achieving the maximized SNM in the ternary logic operation. The $-V_{\text{prg}}$ provided a higher current as well as a longer NTC range in H-MTR, which allowed for a gradual increase in $V_{\text{OUT}}$ value and $V_{\text{IN}}$ range for the intermediate logic state of R-inverter (Supplementary Fig. 8). The $V_{\text{OUT}}$ for the intermediate logic state was successfully controlled to the optimal value of 3 V ($V_{\text{DD}}/2$) with $V_{\text{prg}} = -15$ V and, at this state, the enlarged $V_{\text{IN}}$ range (2 V) of the intermediate logic state, corresponding to one-third of the total range of $V_{\text{IN}}$, was achieved in R-inverter. The DC voltage gain profile is also shown in Fig. 3d, where two distinct gain peaks, namely 1st gain and 2nd gain, are clearly observed from the 1st transition (transition

from logic 2 to logic 1) and 2nd transition (transition from logic 1 to logic 0) of the R-inverter, respectively. With the increasing $|V_{\text{prg}}|$, the 1st gain decreased while the 2nd gain increased due to the gradually increasing $V_{\text{OUT}}$ for the intermediate logic state. Note that only the 1st transition voltage shifted toward the negative direction. This result is related to the asymmetric shift of $V_{\text{TH\_P1}}$ and $V_{\text{TH\_P2}}$ in the H-MTR with $-V_{\text{prg}}$, where $V_{\text{TH\_P1}}$ and $V_{\text{TH\_P2}}$ are responsible for the 1st transition and 2nd transition voltage, respectively. The SNM of the R-inverter for ternary logic operation was examined by plotting the butterfly inverter curve in accordance with $V_{\text{prg}}$ (Fig. 3e and Supplementary Fig. 9). The results revealed a remarkable increase in SNM from 0 V (no margin for the intermediate logic state) at the initial state to 1.25 V (59% of the ideal value) at the optimum programming state ($V_{\text{prg}} = -15$ V). Next, binary logic operation of the R-inverter was analyzed by applying $+V_{\text{prg}}$ to the H-MTR (Fig. 3f). The H-MTR gradually showed the lower current (<1 nA) in NTC region compared to that of the counterpart transistor (>10 nA) with the programming operation, which induced gradual decrease in $V_{\text{OUT}}$ value for the intermediate logic state of R-inverter (Supplementary Fig. 10). At $V_{\text{prg}} = +18$ V, the H-MTR showed a practically off-state current (<100 pA) in NTC region and the intermediate logic state of the R-inverter became not discernible, resulting in the complete conversion to binary inverter. This ternary-to-binary transition was supported by the DC voltage gain profile, at which the 2nd gain decreased continuously and eventually disappeared while the 1st gain increased significantly up to 79 V/V (Fig. 3g). Here, $+V_{\text{prg}}$ does not cause any shift in the 1st and 2nd transition voltages because of the negligible change in $V_{\text{TH\_P1}}$ and $V_{\text{TH\_P2}}$ in the H-MTR. The SNM of the binary logic operation was evaluated with the applied $V_{\text{prg}}$ (Fig. 3h and Supplementary Fig. 11). The SNM for logic 2 and logic 0 (except for the intermediate logic state) was 2.57 V (61% of the ideal value) at the initial state, but the SNM increased up to 3.62 V (85% of the ideal value) as the R-inverter was completely converted into binary-logic-mode by applying $V_{\text{prg}} = +18$ V. The above analysis clearly demonstrates that the key parameters of the R-inverter, such as $V_{\text{OUT}}$ of the intermediate logic state, 1st and 2nd transition voltages, and the gains for each transition, can be modulated systematically by applying $V_{\text{prg}}$ to the H-MTR (Fig. 3i–k).

For practical application, the electrical stability of the R-inverter should also be ensured. The reliability of the R-inverter was examined through cycle endurance test (Fig. 4a). The R-inverter showed three distinct logic states with $V_{\text{OUT}}$ ~ 3 V for the intermediate logic in ternary-logic-mode while exhibited $V_{\text{IN}}$ ~ 3 V for logic 2-to-logic 0 transition voltage without discernable intermediate logic state in binary-logic-mode over 25 cycles of consecutive programming operation, confirming the reliable and reversible binary/ternary logic reconfiguration (Fig. 4b, c). The retention property of the R-inverter was investigated as well, by monitoring the time-dependent variation of the VTCs for ternary logic operation ($V_{\text{prg}} = -15$ V) and binary logic operation ($V_{\text{prg}} = +18$ V) (Fig. 4d, e). For ternary-logic-mode, the R-inverter fully retained $V_{\text{OUT}}$ close to the ideal value (3 V) for the intermediate logic state even after the $10^4$ s. For binary-logic-mode, there was a slight increase in $V_{\text{OUT}}$ and showed 0.3 V for the intermediate logic state after $10^4$ s. Based on the VTCs, the SNM was calculated with respect to the retention time (Fig. 4f). The SNM decreased slightly from 1.02 to 0.91 V for ternary-logic-mode and from 3.6 to 3.48 V for binary-logic-mode after $10^4$ s, both of which correspond to a change less than 9%, confirming the outstanding retention performance for the dynamic circuit.

## Binary/ternary logic conversion-in-memory

The successful demonstration of the R-inverter allowed us to further implement a logic conversion-in-memory, where a series of input signals ($V_{\text{IN}} = 0$ V, $V_{\text{DD}}/2$, and $V_{\text{DD}}$) can produce different combinations of output signals (logic 2, logic 1, and logic 0) depending on the

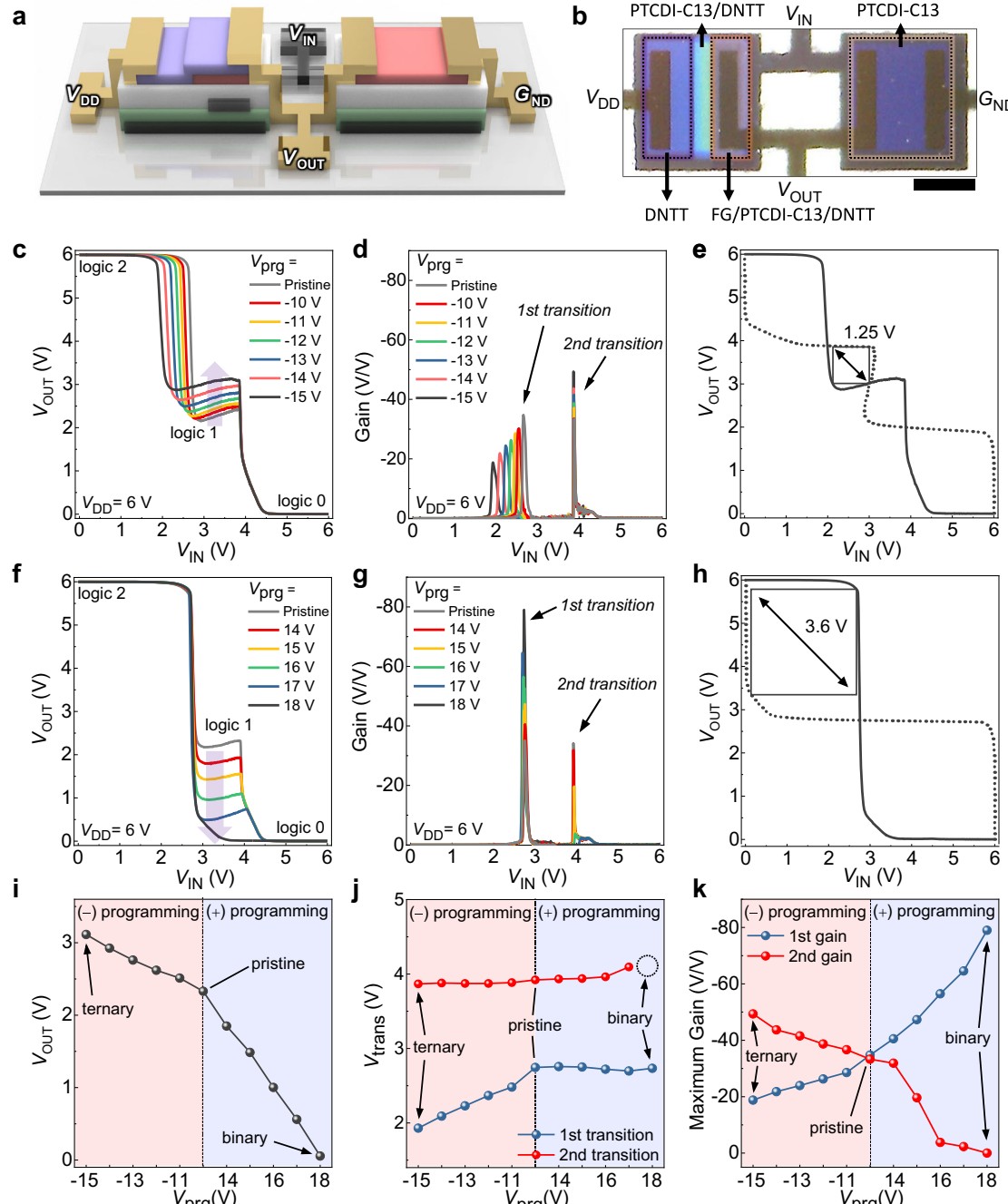

**Fig. 3 | R-inverter. a** A schematic illustration of the R-inverter. **b** An optical microscopy image of the inverter. (scale bar: 500 μm). **c** Voltage transfer characteristics (VTC), and **d** DC gain profiles of the R-inverter with respect to $-V_{prg}$. **e** Butterfly inverter curve of the inverter with ternary operation ($V_{prg} = -15$ V). **f** VTC, and **g** DC gain profiles of the R-inverter with respect to $+V_{prg}$. **h** Butterfly inverter curve of the inverter with binary operation ($V_{prg} = +18$ V). **i** Output voltage ($V_{OUT}$) of intermediate logic state (input voltage ($V_{IN}$) = 3.5 V). **j** 1st and 2nd transition voltages ($V_{trans}$), and **k** 1st and 2nd DC gain values of R-inverter with respect to $V_{prg}$.

memory states of the R-inverter (Fig. 5a). Different from the afore-mentioned R-inverter designed to maximize SNM in binary-logic-mode with a logic 2-to-logic 0 transition voltage at $V_{IN} = V_{DD}/2$, the R-inverter for logic conversion-in-memory should have specific logic for binary-logic-mode as well as ternary-logic-mode at $V_{IN} = V_{DD}/2$, which can be achieved here simply by shifting the 1st and 2nd transition voltage of as-fabricated R-inverter. As mentioned before, these transition voltages are related to the $V_{TH\_P2}$ and $V_{TH\_P1}$ of the H-MTR, respectively. It has been reported that $V_{TH\_P1}$ and $V_{TH\_P2}$ could be controlled by varying the thickness of the p-type and/or n-type semiconductor of the heterojunction transistor because the semi-conductor thickness variation leads to Fermi level shift, and thereby

the change in the charge carrier density[29]. Using this strategy, we could shift $V_{TH\_P1}$ and $V_{TH\_P2}$ in $+V_G$ direction (Supplementary Fig. 12). Accordingly, the corresponding R-inverter represented logic 2 in binary-logic-mode and logic 1 in ternary-logic-mode at $V_{IN} = V_{DD}/2$ (Fig. 5b, c). Therefore, while maintaining the symmetric in-out VTC, the R-inverter could show two different outputs (logic 1 and logic 2) as the intermediate logic depending on the memory state, which enabled the sequential integration of the binary/ternary logic conversion-in-memory. As a proof of concept, two-stage R-inverter was demonstrated (Fig. 5d). The results showed that three different sets of output logic could be implemented according to the memory state of the first and second R-inverters, and the intermediate logic of

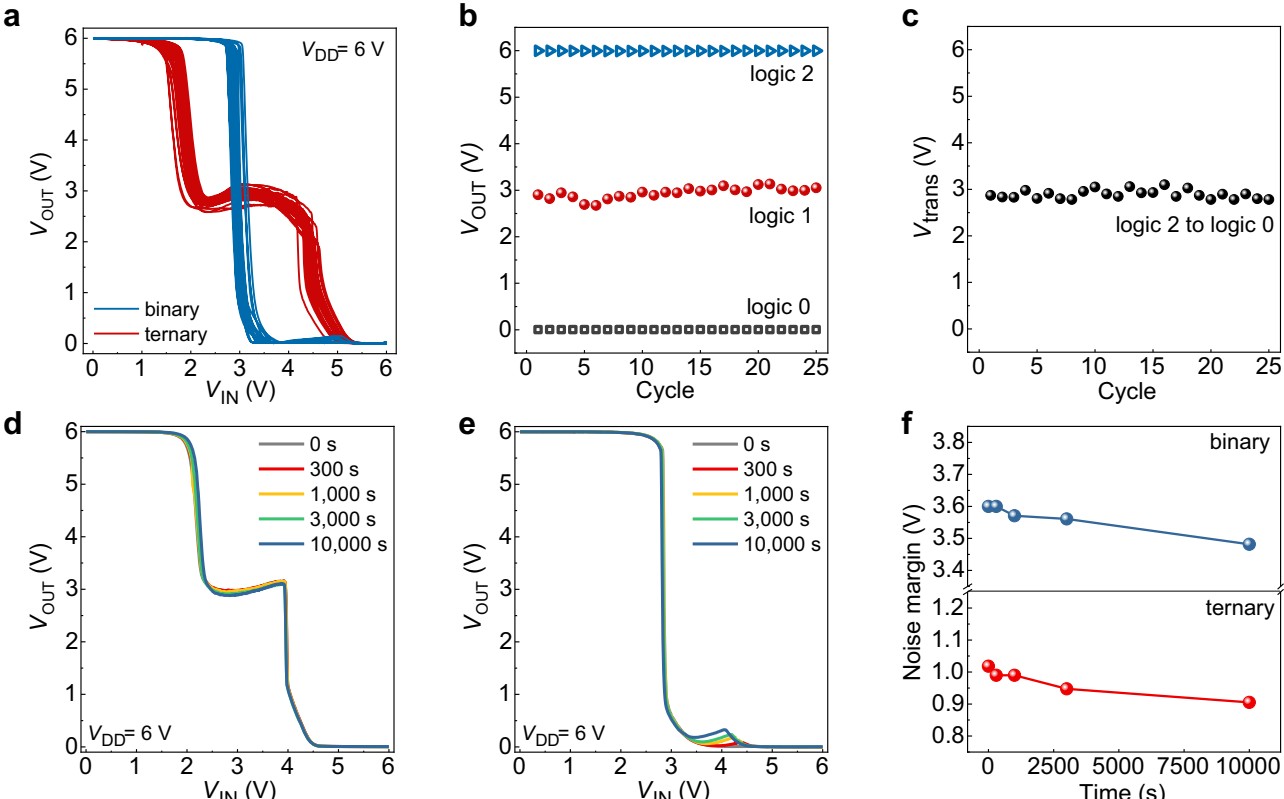

**Fig. 4 | Stability of the R-inverter. a** VTCs of repetitive transition between ternary and binary logic operations up to 25 cycles. **b** Extracted $V_{OUT}$ values at $V_{IN}$ = 0 V, 3 V, and 6 V from each cycle of ternary logic operation. **c** Extracted $V_{tran}$ from each cycle of binary logic operation. **d** The change in VTCs with ternary operation and **e** binary operation vs. time. **f** The change in static noise margin (SNM) of ternary operation and binary operation vs. time.

two-stage R-inverter could be varied to all possible logic values (logic 2, logic 1, and logic 0), providing the functions of STI, PTI and NTI. The dynamic logic output was then monitored through transient measurement (Fig. 5e). The two-stage R-inverter represented hysteresis-free operation throughout the measurement time, and a series of input signal was converted successfully through the first and second R-inverter to all three types of output signals, which was highly correlated with the memory state of each constituent R-inverter, hence proving the dynamic logic conversion-in-memory manipulation. It is important to note that this kind of logic computing method, which directly links binary and ternary logic systems to implement various types of reconfigurable logic gates without involving complex circuits, can only be achieved through the proposed device strategy with dynamic NTC characteristics because the binary/ternary conversion does not result in non-symmetric voltage transfer characteristics as well as a change in $V_{DD}$ (Supplementary Table 3).

Compared to conventional Si-CMOS technology, the proto-type logic devices demonstrated in this work shows relatively higher voltage, lower current, and larger size due to the limited electrical properties of organic materials. However, it should be noted that the proposed device scheme follows conventional CMOS design and does not impose strict limitations on material systems as long as the semiconducting layer can provide NTC characteristics by constructing a reliable p-n heterostructure. Therefore, the performance of the proposed devices can be further improved by using high-*k* inorganic dielectric materials, lithography-compatible semiconductors, or employing other device optimization strategies such as molecular/morphology engineering[65–68], threshold voltage control[69,70], and contact resistance engineering[69,71].

## Discussion

A drain-aligned floating gate heterojunction non-volatile memory transistor was proposed, where the FG is defined only under the drain electrode. The newly developed heterojunction non-volatile memory transistor was capable of controlling the NTC characteristics based on the FG-mediated memory state. The asymmetric configuration of the FG enabled independent modulation of electron injection only from the drain to the channel, thereby allowing the systematic control of the NTC characteristics solely by a simple programming operation, and all the modulation capability was accomplished from a single heterojunction non-volatile memory transistor. Using the programmable NTC characteristics, a binary/ternary reconfigurable logic inverter was implemented successfully. The reconfigurable inverter showed that the range and output values for the intermediate logic state can be modulated according to the memory state of the heterojunction non-volatile memory transistor. Consequently, the reconfigurable inverter was successfully operated in both binary-logic-mode and ternary-logic-mode with a maximized noise margin of 85% and 59% of their corresponding ideal values, respectively. A prototype of binary/ternary logic conversion-in-memory was demonstrated by using two-stage reconfigurable inverter as well, which generated three different sets of output logic signals depending on the memory state of each constituent reconfigurable inverter. Since the exceptionally high noise margin as well as novel logic-conversion circuits demonstrated in this study were achieved through the proposed device architecture rather than unique properties of specific materials, we believe that the proposed scheme will pave the way for multi-functionalization of various logic devices and the implementation of future reconfigurable logic conversion-in-memory based on a variety of material systems.

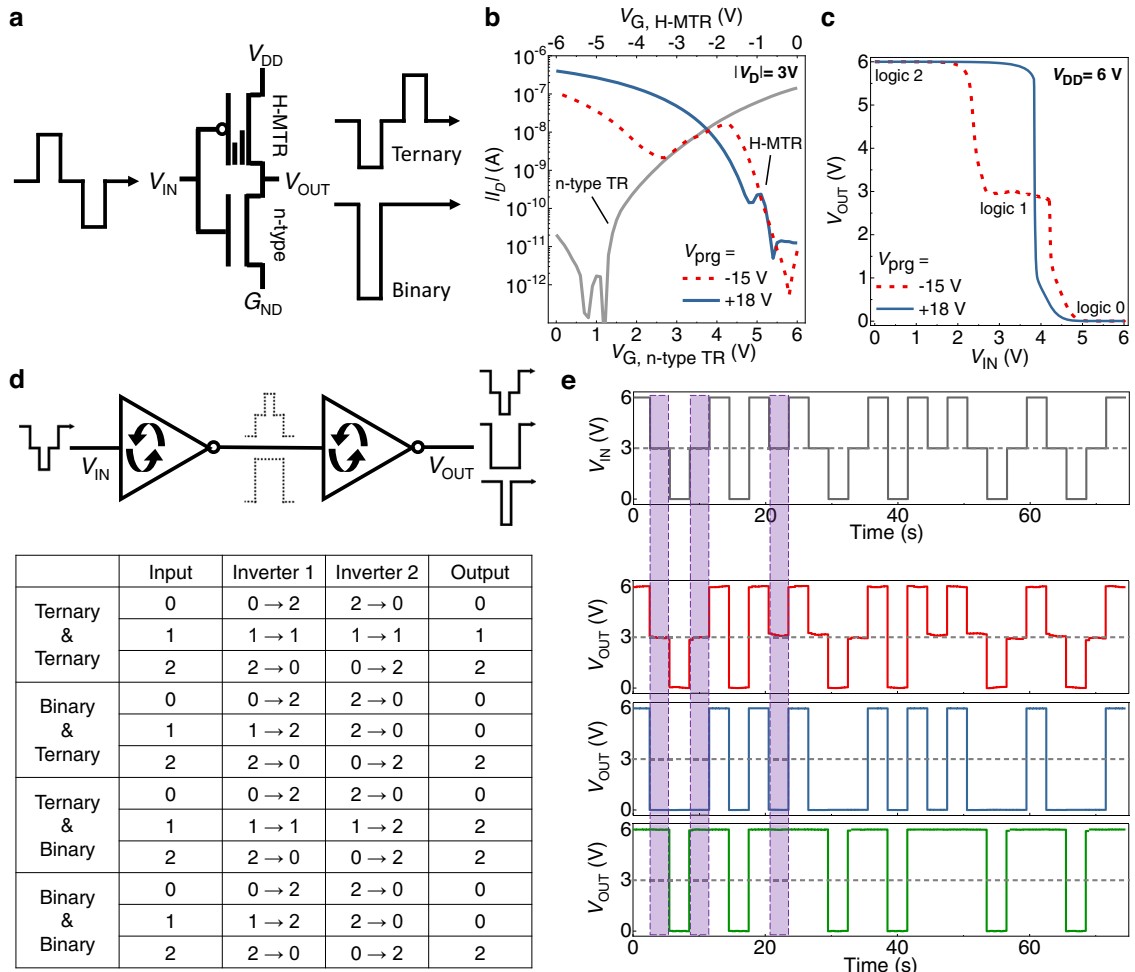

**Fig. 5 | Binary/ternary logic conversion-in-memory. a** A schematic symbol of logic conversion-in-memory. **b** Overlapped transfer characteristics of the H-MTR for $V_{prg} = -15$ V (dotted line) and $V_{prg} = +18$ V (solid line) and n-type transistor. Here, 75 nm-thick DNTT was used. **c** VTCs of the corresponding inverter. $V_{prg} = -15$ V (dotted line) and $V_{prg} = +18$ V (solid line). **d** A schematic symbol of two-stage R-inverter and its truth table. **e** Pulsed measurement of two-stage R-inverter with three different configurations (ternary-ternary, binary-ternary, and ternary-binary/binary-binary).

## Methods

### Materials and substrate

1,3,5-trimethyl-1,3,5-trivinyl cyclotrisiloxane (V3D3 monomer; 95%) was purchased from Gelest, and 2-cyanoethyl acrylate (CEA monomer; >95%), diethylene glycol divinyl ether (DEGDVE monomer; 99%), and *tert*-butyl peroxide (TBPO; 97%) were purchased from Sigma-Aldrich for fabricating the ultrathin polymer dielectric layer. The dinaphtho[2;3-b:2′,3′-f]thieno[3,2-b]thiophene (DNTT; 99%) p-type semi-conductor and N,N′-ditridecyl-perylene-3,4,9,10-tetracarboxylic diimide (PTCDI-C13; 95%) n-type semiconductor were purchased from Sigma-Aldrich as well. All the chemicals were used as received without further purification. Glass substrate of $25 \times 25$ mm was cleaned with deionized (DI) water, acetone, and isopropyl alcohol sequentially for 15 min under ultrasonication, followed by blowing with dry $N_2$ gas.

### Polymer dielectric deposition

Polymer dielectric layers were deposited by an iCVD system. For poly(2-cyanoethyl acrylate-co-diethylene glycol divinyl ether) [p(CEA-co-DEGDVE)] deposition, CEA, DEGDVE and TBPO were vaporized and injected into the iCVD chamber at flow rates of 0.28, 0.28 and 0.48 sccm, respectively. The substrate temperature and chamber pressure were maintained at 30 °C and 60 mTorr, respectively. For poly[1,3,5-trimethyl-1,3,5-trivinyl cyclotrisiloxane] (pV3D3) deposition, V3D3 and TBPO were vaporized and injected into the iCVD chamber at flow rates of 2.5 and 1 standard cubic centimeter per minute (sccm),

respectively. The substrate temperature and chamber pressure were maintained at 40 °C and 300 mTorr, respectively. The filament was heated to 130 °C to decompose the initiator into radicals for both processes.

### Thin film characterization

The thickness of the organic semiconductors and dielectric layer was measured using a spectroscopic ellipsometer (M2000, J. A. Woollam, USA). Information about the band structure of used materials were obtained from the literature[72].

### Device fabrication

For control gate electrodes, floating gate electrodes, and metal–insulator–metal (MIM) devices, 50 nm-thick Al electrodes were thermally deposited at a deposition rate of -1 Å s$^{-1}$. The pCD and pV3D3 polymer dielectric layer was deposited by the iCVD process, and the thickness was measured using an ellipsometer after the deposition. For the n-type semiconductor, PTCDI-C13 was deposited by thermal evaporation at a deposition rate of 0.2–0.3 Å s$^{-1}$ and annealed at 200 °C for 30 min. For the p-type semiconductor, DNTT was deposited at a deposition rate of 0.2–0.3 Å s$^{-1}$. During the deposition, the thickness was monitored in-situ by quartz crystal microbalance (QCM). Then, 70 nm-thick Au source/drain electrodes were thermally deposited at the deposition rate of -0.7 Å s$^{-1}$ through a shadow mask with channel dimensions of 400 ($L$) × 800 ($W$) μm. All

the thermal evaporation was carried out at a chamber pressure lower than $1 \times 10^{-6}$ Torr.

### Device characterization

The electrical properties of all the devices were measured using a B1500A semiconductor device analyzer (Agilent Technologies) in the $N_2$-filled glove box at room temperature.

## Data availability

The data within the article and its Supplementary Information are available from the corresponding authors upon request.

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

## Acknowledgements

This work was supported by the Wearable Platform Materials Technology Center (WMC) funded by the National Research Foundation of Korea (NRF) grant funded by the Korea government (MSIT) (No. NRF-2022R1A5A6000846) (S.G.I.) and also by NRF grant funded by MSIT (2021R1A2B5B03001416, RS-2023-00210194) (S.G.I., H.Y.). This work was also supported by the Technology Innovation Program (1415181712, RS-2022-00144300) (S.G.I.) funded By the Ministry of Trade, Industry & Energy (MOTIE, Korea).

## Author contributions

Chung.L., H.Y., and S.G.I. conceived the idea and designed the experiments. Chung.L. designed, fabricated, and measured all the devices and circuits. Chang.L. assisted device characterization and circuit design. S.L. assisted device fabrication. Chung.L., J.C., H.Y., and S.G.I. wrote the manuscript. All authors reviewed the manuscript and discussed the results.

## Competing interests

The authors declare no competing interests.
