## [Peer Review File · Nature Communications]

REVIEWER COMMENTS

Reviewer #1 (Remarks to the Author):

In this work, the authors presented a heterojunction non-volatile memory transistor featuring a floating gate that enables negative transconductance. Moreover, they investigated the application of such a device in ternary logic providing insightful information in the binary to ternary reconfiguration of two cascaded inverters realized using this technology. The work is quite interesting and valid, but some points need to be addressed by the authors before the reviewer can recommend it for publication on this Journal.

The reviewer is not fully convinced about the degree of novelty of this work with respect to the work already published on Nat. Commun. titled "Vertically stacked, low-voltage organic ternary logic circuits including nonvolatile floating-gate memory transistors" and the one titled "Systematic Control of Negative Transconductance in Organic Heterojunction Transistor for High-Performance, Low-Power Flexible Ternary Logic Circuits" published on Small. The differences in the device architecture and the ternary to binary conversion novel application are clear to the reviewer but the authors should formulate some sound argumentation supporting and explaining the novelty of the proposed work. The reviewer's fear is that if this will not be clearly stated, the proposed work would appear too much as a "follow-up" work and not as a novel enough concept, at least for the considered Journal.

If this concern will be addressed by the authors, some other minor points should be considered:

- line 29 it is not clear what "in unit H-MTR device" means with respect to the formulation of the whole sentence. In general the English writing should be improved.

- line 51-52 "One...characteristics", also here the sentence is not clear from an English formulation point of view.

- It is clear, from the material system perspective and the electrical measurements provided that the proposed device is outperformed by CMOS technology. For a concept device/technology this is perfectly fine, but the end of the scaling era in CMOS technology and the use of FPGA (still based on silicon technology) shouldn't then be used as a motivation for the work, unless a perspective in terms of scalability and improvability of the proposed technology is discussed.

- with respect to this last point, in general the material system employed should be clearly stated from the beginning. It is not clear until page 7-8 what are the materials employed, and this information can not come so late in the paper.

- line 146: can the authors better elaborate on the statement that FN tunneling is the actual phenomenon taking place? Are there references showing this as the actual mechanism? Or why are the authors assuming so?

- as also already briefly commented on, could the authors briefly discuss about voltage scaling perspectives for their devices?

- line 307: fermi  Fermi.

- line 392-395: quite importantly, can the authors comment on the stability of their devices in atmospheric conditions, since the measurements were ran in nitrogen?

Reviewer #2 (Remarks to the Author):

Questions and Comments:

(1) In my opinion FPGAs are a very bad example for reconfigurable electronics as they are very slow and cannot be run-time programmed. I would rather discuss the recent advances of Si based RFETs, which are fully CMOS compatible and show excellent On-currents for both n- and p- type operation.

Mikolajick, T.; Galderisi, G.; Rai, S.; Simon, M.; Böckle, R.; Sistani, M.; Cakirlar, C.; Bhattacharjee, N.; Mauersberger, T.; Heinzig, A.; Kumar, A.; Weber, W. M.; Trommer, J. Reconfigurable Field Effect Transistors: A Technology Enablers Perspective. *Solid-State Electronics* 2022, 194, 108381.

Simon, M.; Liang, B.; Fischer, D.; Knaut, M.; Tahn, A.; Mikolajick, T.; Weber, W. M. Top-Down Fabricated Reconfigurable FET With Two Symmetric and High-Current On-States. *IEEE Electron Device Letters* 2020, 41, 1110–1113.

(2) The introduction is lacking a recognition of state of the art Si and Ge reconfigurable transistors. Especially Ge NW based devices are also capable of providing an NDR mode, which can also be used for MVL.

Sistani, M.; Böckle, R.; Falkensteiner, D.; Luong, M. A.; Den Hertog, M.; Lugstein, A.; Weber, W. Nanometer-Scale Ge-Based Adaptable Transistors Providing Programmable Negative Differential Resistance Enabling Multivalued Logic. *ACS Nano* 15, 18135–18141.

(3) The drive-currents of the device are very low. The authors should provide strategies how to improve this important parameter.

(4) The gate and bias voltage levels of the proposed device concept are fairly high. Can the authors provide ideas how to decrease the applied voltages and enable a reduction of different voltage levels needed for device operation?

(5) What are the actual dimensions of the proposed device? Is the device scalable? Is it compatible with state of the art CMOS fabrication?

(6) For real life applications, it would be of utmost importance to investigate the thermal stability of the device. Can the authors provide data at elevated temperatures? How does the PVR evolve with temperature?

Reviewer #3 (Remarks to the Author):

Authors reported a heterojunction non-volatile memory transistor (H-MTR) containing a drain-aligned floating gate to control the negative transconductance (NTC) characteristics. Finally authors demonstrated ternary/binary dynamic logic conversion-in-memory to generate three different output logic sequences for the same input signal in three logic levels. My comments are follows

1. A drain-aligned floating gate is known and authors claims "newly developed". Please clarify

2. The asymmetric device configuration of H-TR is expected to effectively adjust the amounts of electrons injected into the channel, and thus systematic control of NTC characteristics with programming operation can be achieved. Can authors comment on the p-type and n-type parameters at the drain side? How it affects the NTC characteristics?

3. For non-technical readers, it would be good to have a schematic diagram to explain the concept from device physics to achieving reconfigurable binary/ternary logic conversion.

4. For +ve programming operation, the drain current is confined in one line. Can authors explain this phenomenon?

4.

Response to the Reviewers' Comments

The authors thank the Reviewers for their thorough reading of our manuscript and the valuable comments. We have revised the manuscript and supporting information based on the Reviewers' comments. The Reviewers' comments appear in **black**, and the authors' responses are in blue. In the revised manuscript, **the changes** with respect to the previous version are highlighted.

Reviewer (#1)'s COMMENTS:

In this work, the authors presented a heterojunction non-volatile memory transistor featuring a floating gate that enables negative transconductance. Moreover, they investigated the application of such a device in ternary logic providing insightful information in the binary to ternary reconfiguration of two cascaded inverters realized using this technology. The work is quite interesting and valid, but some points need to be addressed by the authors before the reviewer can recommend it for publication on this Journal.

The reviewer is not fully convinced about the degree of novelty of this work with respect to the work already published on *Nature Communication* titled "Vertically stacked, low-voltage organic ternary logic circuits including nonvolatile floating-gate memory transistors" and the one titled "Systematic Control of Negative Transconductance in Organic Heterojunction Transistor for High-Performance, Low-Power Flexible Ternary Logic Circuits" published on *Small*. The differences in the device architecture and the ternary to binary conversion novel application are clear to the reviewer but the authors should formulate some sound argumentation supporting and explaining the novelty of the proposed work. The reviewer's fear is that if this will not be clearly stated, the proposed work would appear too much as a "follow-up" work and not as a novel enough concept, at least for the considered Journal.

Response:

We appreciate the Reviewer's considerate comments. As the Reviewer #1 pointed out, our group has demonstrated ternary logic circuits in previous publications (*Nat. Commun.*, 2022, 13, 2305 (Ref. 17 in the manuscript) and *Small*, 2021, 17, 2103365 (Ref. 15 in the manuscript)). However, the present work differs totally from our previous reports in that we developed a novel

strategy on device architecture and logic computing method, which have not been presented until now. We would like to highlight the significance of the present work in comparison to our previous reports, as summarized in **Table R1**.

Table R1. Summary of the significance of this work compared to our previous studies.

NTC device	Dynamic NTC control	Logic conversion	Noise margin	Integration level	Applications	Ref
Heterojunction transistor	-	-	-	-	STI	[R1]
Heterojunction transistor	-	-	48 % (Ternary)	-	STI	[R2]
Heterojunction non-volatile memory	0	Binary to ternary	85 % (binary) 59 % (Ternary)	Two-stage inverter	Binary/ternary reconfigurable inverter STI / PTI / NTI Logic conversion-in-memory	This work

STI, standard ternary inverter; PTI, positive ternary inverter; NTI, negative ternary inverter

In our previous study (*Nat. Commun.*, 2022, 13, 2305), a non-volatile memory (flash memory) was introduced as a load transistor and the threshold voltage (V_{TH}) of the load transistor was adjusted to address the conductance (G) mismatch between negative transconductance (NTC) device and load transistor. However, the previous study did not involve any aspects related to the control of NTC characteristics itself, which plays a critical role in defining the intermediate logic state in ternary logic circuits, where a conventional heterojunction transistor was used as-is for NTC device in that study. Another previous study (*Small*, 2021, 17, 2103365) was about the optimization of the conventional heterojunction transistor for G matching between NTC device and load transistor, where NTC characteristics were controlled by the asymmetric width of source/drain electrodes. Similarly, the design of the conventional heterojunction transistor was used as-is for NTC device and the inherent NTC characteristics could not be adjusted once the device layout was fixed.

Compared to previous studies, a completely different type of NTC device is newly developed by uniquely applying a “drain-aligned” floating gate and introducing the non-volatile memory function directly into the heterojunction transistor. Even after device fabrication, the inherent NTC characteristics of the NTC device can still be dynamically controlled by programming operation through the “drain-aligned” floating gate. Therefore, the proposed device enables the dynamic control of the shape and range of NTC characteristics in the ternary logic circuit, which can provide a powerful solution for the unstable intermediate logic state of

the ternary logic inverter derived from G mismatch between NTC device and the load transistor, which has been one of the most challenging issues of multi-valued logic applications. For example, the noise margin of the ternary logic inverter, which is greatly influenced by the intermediate logic state, is improved substantially compared to those from our previous studies. Even compared to previously reported champion ternary logic inverters based on other working principles, the present work shows the highest noise margin values reported to date (**Table R2**).

Table R2. Comparison of static noise margin with the previously reported ternary logic inverters.

Device type	Principle	Materials	Noise margin	Ref
T-CMOS	Band-to-band tunneling	Si	45 %	[R3]
Embedded QDs	Quantized energy level	ZnO QDs	34 %	[R4]
T-CMOS	Threshold switch	Si, NbO ₂	46.5 %	[R5]
Heterojunction transistor	Zero transconductance	PTCDI-C8 / IGZO	-	[R6]
Heterojunction non-volatile memory	Negative transconductance	PTCDI-C13 / DNTT	59 %	This work

T-CMOS, ternary CMOS; QDs, quantum dots;

The proposed device is capable of not only optimizing the intermediate logic state but also utilizing the dynamic NTC characteristics to generate or eliminate the intermediate logic state of the ternary logic inverter simply by turning on and off the NTC characteristics. This enables the device to provide various multi-functional logic gates and logic-in-memory in a ternary logic system. This study also demonstrated logic conversion between binary and ternary systems, which was not possible in previous studies. Furthermore, we achieved the first demonstration of the sequential integration of ternary logic inverters based on NTC devices by implementing a two-stage ternary logic inverter. Depending on the memory state, each inverter can be switched individually and reversibly between binary and ternary logic as well, which can generate three different output logic sequences for the same input signal in three logic levels and therefore can perform all the functions of standard ternary inverter (STI), positive ternary inverter (PTI), and negative ternary inverter (NTI) verified by pulsed measurement (**Fig. R1**).

Figure R1. Binary/ternary logic conversion-in-memory. **a**, Schematic diagram of logic-conversion-in-memory. **b**, Schematic diagram of circuit operation in STI, NTI and PTI. **c**, Pulsed measurement of STI, NTI and PTI.

It is crucial to acknowledge that the dynamic NTC characteristics of the proposed device enables the implementation of various types of reconfigurable logic gates that link binary and ternary logic systems without involving complex circuits. This kind of logic computing method is only achievable through the proposed device strategy because binary/ternary logic conversion does not result in non-symmetric voltage transfer characteristics or require a change in the supply voltage (V_{DD}). To the best of our knowledge, all previously reported binary/ternary logic reconfiguration were accompanied by non-symmetric voltage transfer characteristics or the change in V_{DD} (**Table R3**), which restricts a direct connection between binary and ternary logic gates, thereby limiting flexible circuit operation between binary and ternary logic systems, as well as implementation of multi-functional logic gates and logic conversion-in-memory (**Fig. R2**).

Table R3. Comparison with the previously reported binary/ternary reconfigurable inverter.

Binary logic operation		Ternary logic operation		Integration level	Applications	Ref
V_{IN} / V_{OUT}	V_{DD}	V_{IN} / V_{OUT}	V_{DD}			
1 V / 0.2 V	0.2 V	1 V / 2 V	2 V	-	Binary/ternary reconfigurable inverter	[R7]
30 V / 0.4 V	0.4 V	30 V / 2 V	2 V	-	Binary/ternary reconfigurable inverter	[R8]
40 V / 0.1 V	0.1 V	40 V / 2 V	2 V	-	Binary/ternary reconfigurable inverter	[R9]
60 V / 12 V	26 V	60 V / 26 V	26 V	-	Binary/ternary reconfigurable inverter	[R10]
6 V / 6 V	6 V	6 V / 6 V	6 V	Two-stage inverter	Binary/ternary reconfigurable inverter STI / PTI / NTI Logic conversion-in-memory	This work

**Figure R2. Schematic diagram of logic-conversion circuit. a,** Binary/ternary reconfigurable inverter and corresponding two-stage inverter of previous study and **b,** this work.

Achieving a stable intermediate logic state is currently one of the crucial topics for ternary logic circuit, considering that most of existing ternary logic circuits demonstrated only limited circuit integration levels (e.g., only inverter), which could not be extended to higher integration levels due to the lack of noise margin. In this work, we accomplished a stable intermediate logic state of the ternary logic inverter by developing novel device strategy including the heterojunction non-volatile memory with dynamic NTC characteristics, which is

completely different from previous studies. Beyond simple performance improvements such as higher noise margin and low operating voltage, we demonstrated new logic computing method including a reconfigurable logic inverter with binary/ternary logic conversion capability, direct connection between binary and ternary logic systems, and reconfigurable ternary logic gates using a two-stage reconfigurable inverter as a logic-conversion in memory. To clarify the difference between our present work and previous studies, we revised the manuscript by adding the aforementioned discussion about the novelty of the proposed work.

We revised the manuscript as follows;

(1) Revised the main text in the revised manuscript (p.6)

→ To the best of our knowledge, the SNM of ternary logic operation realized in this study is even higher compared to that of other ternary logic circuits reported previously, which were based on ... and therefore can perform all the functions of a standard ternary inverter (STI), a positive ternary inverter (PTI), and a negative ternary inverter (NTI) according to the corresponding memory states of the constituent R-inverter.

(2) Revised the main text in the revised manuscript (p.17)

→ The results showed that three different sets of output logic could be implemented according to the memory state of the first and second R-inverters, and the intermediate logic of two-stage R-inverter could be varied to all possible logic values (logic “2”, logic “1”, and logic “0”), providing the functions of STI, PTI and NTI.

(3) Revised the main text in the revised manuscript (p.18)

→ It is important to note that this kind of logic computing method, which directly links binary and ternary logic systems to implement various types of reconfigurable logic gates without involving complex circuits, can only be achieved through the proposed device strategy with dynamic NTC characteristics because the binary/ternary conversion does not result in non-symmetric voltage transfer characteristics as well as a change in V_{DD} (Supplementary Table 3).

(4) Added a table comparing the static noise margin of the ternary logic inverters in this work with those of previously reported studies (**Table R2**) to Supplementary Table 2 in Supplementary Information (p. 3)

(5) Added a table comparing binary/ternary reconfigurable inverter of this work with that of previously reported studies (**Table R3**) to Supplementary Table 3 in Supplementary Information (p. 17)

(6) Added the reference [25-29] and [31-34] in the Supplementary Information.

[R1] *Nat. Commun.*, 2022, 13, 2305.

[R2] *Small*, 2021, 17, 2103365.

[R3] *Nat. Electron.*, 2019, 2(7), 307-312.

[R4] *Nat. Commun.*, 2019, 10(1), 1998.

[R5] *IEDM*, 2021, 32.32. 31-32.32. 34.

[R6] *Adv. Mater.*, 2021, 33(29), 2101243.

[R7] *Nat. Nanotechnol.*, 2017, 12(12), 1148-1154.

[R8] *Small*, 2019, 15(11), 1804885.

[R9] *ACS Nano*, 2019, 13(5), 5430-5438.

[R10] *ACS Nano*, 2019, 13(4), 4478-4485.

If this concern will be addressed by the authors, some other minor points should be considered:

Response:

We appreciate the Reviewer #1's constructive and encouraging comments. We present the point-by-point response for each comment as follows.

Q1) line 29 it is not clear what "in unit H-MTR device" means with respect to the formulation of the whole sentence. In general, the English writing should be improved. line 51-52 "One...characteristics", also here the sentence is not clear from a English formulation point of view.

Response:

Thank you very much the Reviewer #1 for pointing out the ambiguous sentences. In the revision process, all the authors thoroughly rechecked the whole manuscript for any other typos or ambiguous expressions. Furthermore, we received English corrections from experts to rectify inappropriate or unclear sentences throughout the manuscript.

The revised parts are as follows;

(1) Revised the main text in the revised manuscript (p.1)

In H-MTR, the transition between “N”-shaped transfer curves with distinct NTC and monolithically current-increasing transfer curves without apparent NTC could be accomplished reliably by programming operation, which is realized in unit H-MTR device.

→ In the H-MTR, a reliable transition between “N”-shaped transfer curves with distinct NTC and monolithically current-increasing transfer curves without apparent NTC can be accomplished through programming operation.

(2) Revised the main text in the revised manuscript (p.3)

Through dynamically modifiable logic functions during circuit operation, the reconfigurable logic can afford more diverse and complex calculations for the given footprint. The key strategy of this circuit operation is to diversify the field-effect characteristics and thus afford multi-functionality into a single transistor unit, which allows the reconfigurable logic in a minimum circuit unit without involving complicated device architecture or hardware cluster configuration.

→ Through the use of dynamically modifiable logic functions during circuit operation, the reconfigurable logic can afford more diverse and complex calculations within a given footprint. The key strategy of this circuit operation is to diversify the field-effect characteristics, thus realizing multi-functionality within a single transistor unit, which allows for reconfigurable logic to be implemented within a minimum circuit unit without involving complicated device architecture or hardware cluster configuration.

(3) Revised the main text in the revised manuscript (p.4)

One of the desirable methods satisfying the transistor with reconfigurable characteristics is integration of memory function to inherent switching function of the transistor by employing charge storage layer, where non-volatile state of the memory can dynamically control the amount of charge carriers (holes for p-type and electrons for n-type) injected into the channel, thereby modifying the magnitude of the electric current at a given gate bias.

→ One desirable method for implementing such reconfigurable transistor is to integrate a memory function into the inherent switching function of the transistor by employing a charge storage layer. In this approach, the non-volatile state of the memory can dynamically control the amount of charge carriers (holes for p-type and electrons for n-type) injected into the channel, thereby modifying the polarity of the transistor.

(4) Revised the main text in the revised manuscript (p.5)

However, it has been quite challenging to adjust the NTC characteristics precisely as an appropriate form to fully exploit the advantages of H-TRs for the target applications. This is because the NTC characteristics are majorly governed by the charge carrier density of two channels in H-TRs, which is under the control of same gate bias.

→ However, it has been quite challenging to adjust the NTC characteristics in a suitable form to fully exploit the advantages of H-TRs for the target applications. This is because the NTC characteristics mainly depend on the charge carrier density of two channels in H-TRs, which is under the control of same gate bias.

(5) Revised the main text in the revised manuscript (p.5)

These issues become more problematic in particular to construct ternary logic circuits, since the magnitude of electric current as well as the range of gate bias for NTC (NTC region) of H-TR need to be manipulated precisely

→ These issues become more problematic when constructing ternary logic circuits, since the magnitude of electric current as well as the range of gate bias for NTC (NTC region) of H-TR need to be manipulated precisely

(6) Revised the main text in the revised manuscript (p.10)

Dynamic NTC characteristics of the H-MTR were investigated while applying negative ($-V_{\text{prg}}$) or positive ($+V_{\text{prg}}$) bias to the CG. For ($-$) programming operation, the electron injection from the drain to channel is promoted and the range of NTC region enlarged gradually as the $|V_{\text{prg}}|$ increases (Fig. 2a).

→ The dynamic NTC characteristics of the H-MTR were investigated by applying negative ($-V_{\text{prg}}$) or positive ($+V_{\text{prg}}$) programming bias to the CG. For ($-$) programming operation, the electron injection from the drain to the channel was promoted and the range of NTC region enlarged gradually as the $|V_{\text{prg}}|$ increased (Fig. 2a).

(7) Revised the main text in the revised manuscript (p.14)

With the increasing $|V_{\text{prg}}|$, the 1st gain decreases while the 2nd gain increases due to the gradually increasing V_{OUT} for the intermediate logic state. Note that V_{IN} shift was detected toward $-V_{\text{IN}}$ direction only for the 1st transition.

→ With the increasing $|V_{\text{prg}}|$, the 1st gain decreased while the 2nd gain increased due to the gradually increasing V_{OUT} for the intermediate logic state. Note that only the 1st transition voltage shifted towards the negative direction.

(8) Revised the main text in the revised manuscript (p.15)

Here, $+V_{\text{prg}}$ does not bring the shift for 1st and 2nd transition voltage because of the negligible change in $V_{\text{TH_pn}}$ and $V_{\text{TH_pp}}$ in the H-MTR.

→ Here, $+V_{\text{prg}}$ does not cause any shift in the 1st and 2nd transition voltages because of the negligible change in $V_{\text{TH_P1}}$ and $V_{\text{TH_P2}}$ in the H-MTR.

Q2) It is clear, from the material system perspective and the electrical measurements provided that the proposed device is outperformed by CMOS technology. For a concept device/technology this is perfectly fine, but the end of the scaling era in CMOS technology and the use of FPGA (still based on silicon technology) shouldn't then be used as a motivation for the work, unless a perspective in terms of scalability and improvability of the proposed technology is discussed. With respect to this last point, in general the material system employed should be clearly stated

from the beginning. It is not clear until page 7-8 what are the materials employed, and this information cannot come so late in the paper.

Response:

We appreciate the Reviewer #1 for bringing to our attention about the over-estimated expressions in the original manuscript. We fully agree that terms like "end of the scaling era" and "FPGA" were not appropriate and did not accurately reflect the significance of our work. As the Reviewer #1 pointed out, the real value of our study lies in the novel concept of devices that have the potential to increase the computational efficiency and integration density of logic circuits in a given area. Our stance was to emphasize the need for concurrent research and advancement in such novel device architecture to circumvent the device size scaling issue. Regarding the material system, we used organic small molecules, dinaphtho[2,3-b:2',3'-f]thieno[3,2-b]thiophene (DNNT) and *N,N'*-ditridecyl-3,4,9,10-perylenetetracarboxylic diimide (PTCDI-C13) for p-type and n-type semiconductors, respectively, to evaluate the feasibility of the proposed device, since they have shown stable NTC characteristics when p-n heterojunction is formed under laboratory conditions^[R11]. However, the proposed device scheme does not impose strict limitations on material systems, as long as the semiconducting layer can produce NTC characteristics by constructing reliable p-n heterostructure.

Following the Reviewer #1's suggestion, we revised the manuscript by alleviating our claims to the scope of our model study and adding the information about the material system we used at the beginning of the manuscript.

We revised the manuscript as follows;

(1) Revised the main text in the revised manuscript (p.3)

→ As visual/speech recognition, smart healthcare systems, and other personalized artificial intelligence (AI) technologies become ubiquitous in daily life, the demand for ... limiting the integration density and information processing capability of organic ICs¹². Reconfigurable electronics can be a promising breakthrough for such scaling issues¹³.

(2) Revised the main text in the revised manuscript (p.4)

→ Several reconfigurable logic devices have been recently implemented using various device architectures and state-of-the-art material systems, including Si^{14,15}, Ge^{16,17}, transition metal dichalcogenides (TMDs)¹⁸⁻²² ... the amount of charge carriers (holes for p-type and electrons for n-type) injected into the channel, thereby modifying the polarity of the transistor²⁰.

(3) Revised the main text in the revised manuscript (p.18)

→ Compared to conventional Si-CMOS technology, the proto-type logic devices demonstrated in this work shows relatively higher voltage, lower current, and larger size ... by using high-*k* inorganic dielectric materials, lithography-compatible semiconductors, or employing other device optimization strategies such as molecular/morphology engineering⁶⁵⁻⁶⁸, threshold voltage control^{69,70}, and contact resistance engineering^{69,71}.

(4) Revised the main text in the revised manuscript (p.19)

→ Since the exceptionally high noise margin as well as novel logic-conversion circuits demonstrated in this study were achieved through the proposed ... for multi-functionalization of various logic devices and the implementation of future reconfigurable logic conversion-in-memory based on a variety of material systems.

(5) Added the reference [1-22] and [65-71] in the revised manuscript.

[R11] *Adv. Mater.*, 2019, 31(29), 1808265.

Q3) line 146: can the authors better elaborate on the statement that FN tunneling is the actual phenomenon taking place? Are there references showing this as the actual mechanism? Or why are the authors assuming so?

Response:

For tunneling dielectric layer (TDL), we used poly(1,3,5-trivinyl-1,3,5-trimethyl cyclotrisiloxane) (pV3D3), which is deposited by a vapor-phase polymer deposition process, termed initiated chemical vapor deposition (iCVD). The Fowler-Nordheim (F-N)-like tunneling phenomenon of this material has already been investigated in our previous study^[R12] by analyzing the current densities (J) – electric field (E) characteristics of metal/insulator/metal (MIM) device, where J – E characteristics closely overlapped over a temperature ranging from $-140\text{ }^{\circ}\text{C}$ to $10\text{ }^{\circ}\text{C}$ indicating temperature-independent carrier conduction. Also, a plot of $\ln |J/E^2|$ versus $1/E$ of pV3D3 MIM device clearly showed an apparent linear curve with good fit to the F-N tunneling equation^[R13] at high E . We confirmed that the J – E characteristics and $|J/E^2|$ – $1/E$ characteristics were reproduced clearly in this study, which are consistent to our previous report (Fig. R3). In addition, the outstanding retention characteristics of the memory devices with pV3D3 TDL in this study and other references^[R14, R15] also strongly support that the charge traps occurred by F-N-like tunneling mechanism through pV3D3 TDL.

Figure R3. Electrical characteristics of the Al/pV3D3/Al MIM device. a, The J – E and b, $\ln |J_i / E_i^2| - 1 / E_i$ characteristics.

We revised the manuscript by adding the above discussion about F-N-like tunneling phenomenon of pV3D3 as follows;

(1) Revised the main text in the revised manuscript (p.8)

→ This enables a higher electric field (E) to be applied mostly across the TDL than the BDL during the programming operation, facilitating the Fowler-Nordheim (F-N)-like

tunneling through the TDL⁵⁸ while minimizing the charge leakage through the BDL (Supplementary Fig. 1).

(2) Added the $\ln |J/E^2|$ versus $1/E$ of characteristics of pV3D3 MIM device (Fig. R3b) to Supplementary Figure 1 in Supplementary Information (p. 4)

(3) More detailed discussion for F-N-like tunneling phenomenon of the pV3D3 and the related references were added in Supplementary Information (p. 4)

→ For tunneling dielectric layer (TDL), we used poly(1,3,5-trivinyl-1,3,5-trimethyl cyclotrisiloxane) (pV3D3) which can be deposited by a vapor-phase polymer deposition process, termed initiated chemical vapor deposition (iCVD). The pV3D3 TDL showed the Fowler-Nordheim (F-N)-like tunneling behavior in the E range from ~ 4 MV/cm (Supplementary Fig. 1b, c), which is fully consistent with our previous report²⁹.

(4) Added the reference [58] in the revised manuscript.

(5) Added the reference [29] in the Supplementary Information.

[R12] *Nat. Mater.*, 2015, 14(6), 628-635.

[R13] *J. Appl. Phys.*, 1969, 40(1), 278-283.

[R14] *Nat. Commun.*, 2017, 8(1), 725.

[R15] *Adv. Funct. Mater.*, 2017, 27(43), 1703545.

Q4) As also already briefly commented on, could the authors briefly discuss voltage scaling perspectives for their devices?

Response:

The device operating voltage can be lowered by increasing capacitance per unit area (C_i) of the gate dielectric, which improves the switching characteristics of the transistors with high driving current as well as low subthreshold swing. We already employed polymer gate dielectric called poly(2-cyanoethyl acrylate-co-diethylene glycol divinyl ether) [p(CEA-co-DEGDVE)]

which has high dielectric constant ($k > 6$)^[16]. However, if inorganic gate dielectrics such as HfO₂ with much higher dielectric constant ($k > 20$) are employed, we believe further voltage scaling can be achieved.

Also, the V_{TH} of our device is relatively high (1.5 V for pull-up heterojunction non-volatile memory transistor and 2 V for pull-down n-type transistor). There are several methods to effectively control the V_{TH} for organic thin-film transistors (OTFTs) such as adding dopants to the semiconducting layer by co-evaporation^[R17] or creating an electrostatic potential via surface modification of the gate dielectric layer^[R18], both of which allow to adjust the hole/electron density within the semiconductor.

Besides, OTFTs typically have high contact resistance due to van der Waals (vdW) metal-molecule contact, which induces decoupling of electronic states between metals and molecules^[R19]. Therefore, reducing the contact resistance of OTFTs by contact doping^[R17] or inserting charge injection layer between organic semiconductor and contact metal^[R20] is one of the effective ways to lower operating voltage.

Along with the contact resistance, the charge transport characteristics of the OTFTs are mostly limited by low crystallinity and numerous grain boundaries of the organic semiconductors, which induces charge trapping and low intrinsic mobility of the channel^[R21]. Molecular engineering^[R22, R23], surface modification^[R24], and advancements in deposition technology^[R25] can be employed to control the molecular orientation and achieve highly aligned molecular packing structures of the organic semiconductors, thereby further increasing the intrinsic mobility of the channel and lowering the operating voltage of the device.

Additionally, as already mentioned in the manuscript, programming voltage can be further reduced by allowing a greater amount of programming voltage to be applied to the floating gate of the device, which is related to gate coupling ratio (α_{CR}) defined as following equation^[R26]

$$\alpha_{CR} = V_{FG} / V_G = (C_{BDL} / C_{BDL} + C_{TDL})$$

where V_{FG} is the amount of voltage applied to floating gate, C_{BDL} is the capacitance of blocking dielectric and C_{TDL} is the capacitance of tunneling dielectric. Therefore, by lowering C_{TDL} and increasing C_{BDL} , that is, by reducing the k of the tunneling dielectric layer and increasing the k

of the blocking dielectric layer, the α_{CR} can be maximized and the programming voltage can be effectively reduced accordingly.

We revised the manuscript by adding the above discussion about the voltage scaling of the proposed device as follows;

(1) Revised the main text in the revised manuscript (p. 3)

→ Over the past two decades, there have been numerous intensive efforts in material engineering or device optimization to boost the performance of OTFTs⁷⁻¹¹.

(2) Revised the main text in the revised manuscript (p.18)

→ Compared to conventional Si-CMOS technology, the proto-type logic devices demonstrated in this work shows relatively higher voltage, lower current, and larger size ... by using high- k inorganic dielectric materials, lithography-compatible semiconductors, or employing other device optimization strategies such as molecular/morphology engineering⁶⁵⁻⁶⁸, threshold voltage control^{69,70}, and contact resistance engineering^{69,71}.

(3) Added the reference [7-11] and [65-71] in the revised manuscript.

[R16] *ACS Appl. Mater. Interfaces*, 2017, 9(24), 20808-20817.

[R17] *Chem. Rev.*, 2016, 116(22), 13714-13751.

[R18] *Adv. Funct. Mater.*, 2016, 26(36), 6574-6582

[R19] Sze, S. M., Li, Y., & Ng, K. K., 2021, *Physics of semiconductor devices*. John wiley & sons.

[R20] *Adv. Mater.*, 2022, 34(2), 2104075.

[R21] *Chem. Soc. Rev.*, 2010, 39(7), 2643-2666.

[R22] *Nat. Rev. Chem.*, 2020, 4(2), 66-77.

[R23] *Adv. Funct. Mater.*, 2022, 32(21), 2200843.

[R24] *Adv. Mater.*, 2010, 22(34), 3857-3875.

[R25] *Energy Environ. Sci.*, 2014, 7(7), 2145-2159.

[R26] Nat. Commun. 2017, 8, 725 (2017).

Q5) line 307: fermi  Fermi.

Response:

We appreciate the Reviewer #1's attentive review. We also thoroughly rechecked the whole manuscript and corrected all errors and typos as much as we can. We revised the manuscript as follows;

(1) Revised the main text in the revised manuscript (p.17)

It was also reported that V_{TH_pp} and V_{TH_pn} could be controlled by varying the thickness of the p-type and/or n-type semiconductor of the heterojunction transistor because the semiconductor thickness variation leads to fermi level shift, and thereby the change in the charge carrier density¹⁵.

→ It **has been** reported that V_{TH_P1} and V_{TH_P2} could be controlled by varying the thickness of the p-type and/or n-type semiconductor of the heterojunction transistor because the semiconductor thickness variation leads to **Fermi** level shift, and thereby the change in the charge carrier density²⁹.

(2) Revised the main text in the revised manuscript (p.4)

Compared to conventional digital system, the MVL system uses more than three logic states, and thereby is capable of providing higher data processing efficiency with enhanced integration density in the same design rule.

→ Compared to conventional digital **systems**, the MVL **systems use** more than three logic states, **enabling** higher data processing efficiency with enhanced integration density **within** the same design rule.

(3) Revised the main text in the revised manuscript (p.5)

However, most of the previously reported studies have showed non-symmetric in-out voltage transfer characteristics (VTC) for a ternary logic inverter due to uncontrolled NTC characteristics,

→ However, most of the previously reported studies have shown non-symmetric in-out voltage transfer characteristics (VTC) for a ternary logic inverter due to uncontrolled NTC characteristics,

(4) Revised the main text in the revised manuscript (p.10)

Compared to the I_{Valley} which was practically constant with the value of about 0.6 nA, the I_{Peak} gradually increased from 7 nA (pristine) to the 19 nA ($V_{\text{prg}} = -15$ V) with the higher V_{prg} , leading to the larger peak-to-valley current ratio.

→ Compared to the I_{Valley} which was practically constant with a value of about 0.6 nA, the I_{Peak} gradually increased from 7 nA (pristine) to the 19 nA ($V_{\text{prg}} = -15$ V) with higher V_{prg} , leading to the larger peak-to-valley current ratio.

(5) Revised the main text in the revised manuscript (p.12)

This allowed the H-MTR to maintain its off-state below the $V_{\text{TH_pp}}$ and to operate as if it is a monolithically current-increasing transistor without the NTC, which can lead binary logic operation without distinct intermediate logic state when integrated to inverter.

→ This allowed the H-MTR to maintain its off-state below the $V_{\text{TH_p2}}$ and to operate as if it is a monolithically current-increasing transistor without the NTC, which can lead to binary logic operation without distinct intermediate logic state when integrated into an inverter.

(6) Revised the main text in the revised manuscript (p.17)

The two-stage R-inverter represented hysteresis-free operation throughout the measurement time, and a series of input signal was converted successfully through the 1st and 2nd R-inverter to all three types of output signals, which is highly correlated with the memory state of each constituent R-inverter, hence proving the dynamic logic conversion-in-memory manipulation.

→ The two-stage R-inverter represented hysteresis-free operation throughout the measurement time, and a series of input signal was converted successfully through the first and second R-inverter to all three types of output signals, which was highly

correlated with the memory state of each constituent R-inverter, hence proving the dynamic logic conversion-in-memory manipulation.

Q6) line 392-395: quite importantly, can the authors comment on the stability of their devices in atmospheric conditions, since the measurements were run in nitrogen?

Response:

As noted by the Reviewer #1, all electrical measurements were performed in a nitrogen environment because organic semiconductors are often unstable in air ambient. In this study, dinaphtho[2,3-b:2',3'-f]thieno[3,2-b]thiophene (DNNT) and N,N'-ditridecyl-3,4,9,10-perylene-tetracarboxylic diimide (PTCDI-C13) were used for p-type and n-type organic semiconductor, respectively. To investigate air stability, DNNT and PTCDI-C13 organic thin-film transistors (OTFTs) and their heterojunction non-volatile memory transistor (H-MTR) characteristics were measured under ambient condition (20 °C, 45% relative humidity, RH) (**Fig. R4**). DNNT OTFT was quite stable in air ambient (**Fig. R4a**), whereas PTCDI-C13 OTFT degraded readily when exposed to air and showed a substantial decrease in mobility and V_{TH} shift with increased air exposure time (**Fig. R4b**), which is consistent with previous reports^[R27, R28]. As a result, DNNT/PTCDI-C13 heterojunction transistor also showed significant performance variations in the air (**Fig. R4c**).

However, by employing thin-film encapsulation layer, the air stability issue of organic semiconductors can be mitigated substantially. We encapsulated the devices with 20 nm-thick Al₂O₃ layer via atomic layer deposition (ALD) process. The initial electrical properties of the Al₂O₃-encapsulated DNNT and PTCDI-C13 OTFTs and DNNT/PTCDI-C13 heterojunction transistor were fully maintained during the entire measurement period of approximately 144 hours, consistent with our previous report^[R29]. (**Fig. R5**)

We revised the manuscript by adding the above discussion on the air stability aspect as follows;

(1) Revised the main text in the revised manuscript (p.12)

→ The air stability and thermal stability of the H-MTRs were further investigated to

examine their practical applicability for logic devices. The H-MTRs ... electrical properties of the Al₂O₃-encapsulated H-MTRs were fully maintained during the entire measurement period of approximately 144 hours (Supplementary Fig. 6).

(2) Added the air stability of the Al₂O₃ encapsulated H-MTR (Fig. R5c) to Supplementary Figure 6 in Supplementary Information (p. 10)

Figure R4. Air stability of the devices without encapsulation layer. **a**, The transfer characteristics of DNTT OTFT, **b**, PTCDI-C13 OTFT, and **c**, H-MTR with respect to air exposure time.

Figure R5. Air stability of the devices with encapsulation layer. a, The transfer characteristics of DNTT OTFT, **b,** PTCDI-C13 OTFT, and **c,** H-MTR with respect to air exposure time.

[R27] *Electron. Mater. Lett.*, 2019, 15, 166-170.

[R28] *Phys Status Solidi Rapid Res Lett.*, 2013, 7(7), 469-472.

[R29] *Adv. Electron. Mater.*, 2016, 2, 1500385

Reviewer (#2)'s COMMENTS:

Q1) In my opinion FPGAs are a very bad example for reconfigurable electronics as they are very slow and cannot be run-time programmed. I would rather discuss the recent advances of Si based RFETs, which are fully CMOS compatible and show excellent On-currents for both n- and p-type operation.

Mikolajick, T.; Galderisi, G.; Rai, S.; Simon, M.; Böckle, R.; Sistani, M.; Cakirlar, C.; Bhattacharjee, N.; Mauersberger, T.; Heinzig, A.; Kumar, A.; Weber, W. M.; Trommer, J. Reconfigurable Field Effect Transistors: A Technology Enablers Perspective. *Solid-State Electronics* 2022, 194, 108381.

Simon, M.; Liang, B.; Fischer, D.; Knaut, M.; Tahn, A.; Mikolajick, T.; Weber, W. M. Top-Down Fabricated Reconfigurable FET With Two Symmetric and High-Current On-States. *IEEE Electron Device Letters* 2020, 41, 1110–1113.

Response:

We would like to express our deep gratitude to the Reviewer #2 for giving us the constructive comments. In our manuscript, we cited FPGAs as an example of reconfigurable electronics, as they allow for various logic gate functions through programming operations, much like other reconfigurable devices. However, we fully agree with the Reviewer #2 in that FPGAs are slow and cannot be programmed at runtime. As constructively suggested by Reviewer #2, we have revised the manuscript's introduction by replacing the example of FPGAs with state-of-the-art reconfigurable field-effect transistors that employ various material systems, including Si. The revised introduction is as follows:

(1) Revised the main text in the revised manuscript (p.3)

→ Reconfigurable electronics can be a promising breakthrough for such scaling issues¹³.

(2) Revised the main text in the revised manuscript (p.4)

→ Several reconfigurable logic devices have been recently implemented using various device architectures and state-of-the-art material systems, including Si^{14,15}, Ge^{16,17},

transition metal dichalcogenides (TMDs)¹⁸⁻²², whose polarity can be dynamically toggled either to p-type or n-type.

(3) Added the reference [13-22] in the revised manuscript.

Q2) The introduction is lacking a recognition of state-of-the-art Si and Ge reconfigurable transistors. Especially Ge NW based devices are also capable of providing an NDR mode, which can also be used for MVL.

Sistani, M.; Böckle, R.; Falkensteiner, D.; Luong, M. A.; Den Hertog, M.; Lugstein, A.; Weber, W. Nanometer-Scale Ge-Based Adaptable Transistors Providing Programmable Negative Differential Resistance Enabling Multivalued Logic. *ACS Nano* 15, 18135–18141.

Response:

Thanks to the constructive comment from the Reviewer #2, we became aware of the state-of-the-art reconfigurable transistors that utilize Si and Ge, and can be employed for multi-valued logic (MVL) through negative differential resistance (NDR). We added references in the revised manuscript to support our statement on reconfigurable electronics and MVL, and are grateful for the suggestion.

(1) Revised the main text in the revised manuscript (p.4)

→ Several reconfigurable logic devices have been recently implemented using various device architectures and state-of-the-art material systems, including Si^{14,15}, Ge^{16,17}, transition metal dichalcogenides (TMDs)¹⁸⁻²², whose polarity can be dynamically toggled either to p-type or n-type.

(2) Revised the main text in the revised manuscript (p.4)

→ Along with the reconfigurable logic, multi-valued logic (MVL) has emerged as a promising approach for data-intensive applications^{16,24}.

(3) Added the reference [14-22] and [24] in the revised manuscript.

Q3) The drive-currents of the device are very low. The authors should provide strategies how to improve this important parameter.

Response:

We are grateful for the valuable feedback provided by Reviewer #2. In thin-film transistors, the driving currents can be expressed by the following equation

$$I_{DS} = C_i \mu_{sat} (W / 2L) (V_G - V_{TH})^2$$

where C_i is the capacitance per unit area, μ_{sat} is the saturation mobility, W and L are the channel width and length, respectively, and V_G and V_{TH} are the applied gate voltage and threshold voltage, respectively. Therefore, the driving currents of the device can be improved by introducing (1) gate dielectric with high C_i , (2) channel materials with higher μ_{sat} , (3) short channel length.

For about strategy (1), we already used polymer dielectric called poly(2-cyanoethyl acrylate-co-diethylene glycol divinyl ether) [p(CEA-co-DEGDVE)] with high dielectric constant (k) ($k > 6$)^[R30] as blocking dielectric layer to achieve low-voltage operation of the proposed device. However, the value of k of the p(CEA-co-DEGDVE) is outperformed by high- k inorganic dielectrics such as Al_2O_3 ($k > 10$), HfO_2 ($k > 20$). Also, the thickness of the p(CEA-co-DEGDVE) blocking dielectric in this work (> 70 nm) was far thicker than the thickness of blocking dielectric layer of silicon-oxide-nitride-oxide-silicon (SONOS) flash memory (< 10 nm)^[R31], limiting the C_i . Therefore, by replacing the blocking dielectric layer with high- k inorganic dielectrics and reducing the thickness of the blocking dielectric layer, the C_i can be maximized.

In case of strategy (2), the charge transport characteristics of the OTFTs are mostly limited by low crystallinity and numerous grain boundaries of the organic semiconductors, which induces charge trapping and low intrinsic mobility of the channel^[R32]. Molecular engineering^[R33, R34], surface modification^[R35], and advancements in deposition technology^[R36] can be employed to control the molecular orientation and achieve highly aligned molecular packing structures of the organic semiconductors, thereby further increasing the intrinsic mobility of the channel and lowering the operating voltage of the device.

Unfortunately, applying the (3) strategy to the current material system of this work is not straightforward because there is a limitation to directly apply conventional photo-lithography to organic materials, due to their high susceptibility to high temperature and solvent. However, proposed device scheme does not impose strict limitations on material systems as long as the semiconducting layer can produce NTC characteristics by constructing reliable p-n heterostructure. It has been reported that heterojunction of 2D transition metal dichalcogenides, metal-oxide semiconductors or their combinations, all of which are fully lithography-compatible, can also realize NTC characteristics^[R37-39]. Therefore, strategy (3) is still applicable if lithography-compatible semiconductors are employed, and the driving currents of the device will be significantly increased with channel length scaling.

We revised the manuscript by adding the above discussion about the strategies to increase driving current as follows;

(1) Revised the main text in the revised manuscript (p. 3)

→ Over the past two decades, there have been numerous intensive efforts in material engineering or device optimization to boost the performance of OTFTs⁷⁻¹¹.

(2) Revised the main text in the revised manuscript (p.18)

→ Compared to conventional Si-CMOS technology, the proto-type logic devices demonstrated in this work shows relatively higher voltage, lower current, and larger size ... by using high-*k* inorganic dielectric materials, lithography-compatible semiconductors, or employing other device optimization strategies such as molecular/morphology engineering⁶⁵⁻⁶⁸, threshold voltage control^{69,70}, and contact resistance engineering^{69,71}.

(3) Added the reference [7-11] and [65-71] in the revised manuscript.

[R30] *ACS Appl. Mater. Interfaces*, 2017, 9(24), 20808-20817.

[R31] *IEEE Circuits Syst. Mag.*, 2000, 16(4), 22-31.

[R32] *Chem. Soc. Rev.*, 2010, 39(7), 2643-2666.

- [R33] *Nat. Rev. Chem.*, 2020, 4(2), 66-77.
[R34] *Adv. Funct. Mater.*, 2022, 32(21), 2200843.
[R35] *Adv. Mater.*, 2010, 22(34), 3857-3875.
[R36] *Energy Environ. Sci.*, 2014, 7(7), 2145-2159.
[R37] *Nanoscale*, 2019, 11(11), 4701-4706.
[R38] *Appl. Phys. Lett.*, 2014, 105(21), 213507.
[R39] *J. Appl. Phys.*, 2017, 121(12), 124504.

Q4) The gate and bias voltage levels of the proposed device concept are fairly high. Can the authors provide ideas how to decrease the applied voltages and enable a reduction of different voltage levels needed for device operation?

Response:

Thank you for the Reviewer's considerate comments. In analogy to the answer for Q3, the operating voltage can be reduced by increasing the C_i of the gate dielectric, which in turn improves the switching characteristics of the transistors with high driving current and low subthreshold swing.

Also, the V_{TH} of our device is relatively high (1.5 V for pull-up heterojunction non-volatile memory transistor and 2 V for pull-down n-type transistor). There are several methods to effectively control the V_{TH} for organic thin-film transistors (OTFTs) such as adding dopants to the semiconducting layer by co-evaporation^[R40] or creating an electrostatic potential via surface modification of the gate dielectric layer^[R41], both of which allow to adjust the hole/electron density within the semiconductor.

Besides, OTFTs typically have high contact resistance due to the van der Waals (vdW) metal-molecule contact, which induces decoupling of electronic states between metals and molecules^[R42]. Therefore, reducing the contact resistance of OTFTs by contact doping^[40] or inserting charge injection layer between organic semiconductor and contact metal^[R43] is one of the effective ways to lower the operating voltage.

Additionally, as already mentioned in the manuscript, programming voltage can be lowered by designing the gate dielectric configuration to apply greater amount of programming voltage to the floating gate, which is related to gate coupling ratio (α_{CR}) defined as following

equation^[R44]

$$\alpha_{CR} = V_{FG} / V_G = (C_{BDL} / C_{BDL} + C_{TDL})$$

where V_{FG} is the amount of voltage applied to floating gate, C_{BDL} is the capacitance of blocking dielectric and C_{TDL} is the capacitance of tunneling dielectric. Therefore, by lowering C_{TDL} and increasing C_{BDL} , that is, by reducing the k of the tunneling dielectric layer and increasing the k of the blocking dielectric layer, the α_{CR} can be maximized and the programming voltage can be effectively lowered.

The primary goal of this work is to introduce a new device architecture that provides a novel logic computing method and thus, has the potential to enhance the computational efficiency and integration density of next-generation logic circuits. Therefore, in this study, we mainly focused on evaluating the feasibility of the proposed devices. Nevertheless, there are still opportunities that can improve the performance of the proposed device with the use of the aforementioned strategies.

We revised the manuscript by adding the above discussion about the methods to reduce operating voltage as follows;

(1) Revised the main text in the revised manuscript (p. 3)

→ Over the past two decades, there have been numerous intensive efforts in material engineering or device optimization to boost the performance of OTFTs⁷⁻¹¹.

(2) Revised the main text in the revised manuscript (p.18)

→ Compared to conventional Si-CMOS technology, the proto-type logic devices demonstrated in this work shows relatively higher voltage, lower current, and larger size ... by using high- k inorganic dielectric materials, lithography-compatible semiconductors, or employing other device optimization strategies such as molecular/morphology engineering⁶⁵⁻⁶⁸, threshold voltage control^{69,70}, and contact resistance engineering^{69,71}.

(3) Added the reference [7-11] and [65-71] in the revised manuscript.

[R40] *Chem. Rev.*, 2016, 116(22), 13714-13751.

[R41] *Adv. Funct. Mater.*, 2016, 26(36), 6574-6582

[R42] Sze, S. M., Li, Y., & Ng, K. K., 2021, *Physics of semiconductor devices*. John wiley & sons.

[R43] *Adv. Mater.*, 2022, 34(2), 2104075.

[R44] *Nat. Commun.* 2017, 8, 725 (2017).

Q5) What are the actual dimensions of the proposed device? Is the device scalable? Is it compatible with state-of-the-art CMOS fabrication?

Response:

Fig. R6 shows schematic diagram of a binary-ternary logic reconfigurable inverter and its corresponding shadow mask layout. Each inverter has a size of 3,000 μm by 5,000 μm . The channel length and width of the heterojunction transistor correspond to 500 μm and 1,000 μm , respectively. The source and drain electrodes are equally spaced at 250 μm from the p-n junction edge. The drain-aligned floating gate is located only below the drain electrode area with a gap of 150 μm from the p-n junction edge.

Figure R6. Dimensions of the binary/ternary reconfigurable logic inverter. a, Schematic diagram, **b,** An optical microscopy image (scale bar: 500 μm), and **c,** Shadow mask layout of the binary-ternary reconfigurable logic inverter (scale bar: 500 μm).

We fully acknowledge the Reviewer's concern about the scalability of the proposed device, given its considerably large scale. The channel materials used in this study are small molecular organic semiconductors, which are compatible with a vapor-phase thermal deposition process and can be patterned using the shadow masks. Also, each small molecular organic semiconductor has intrinsic p-type and n-type properties, making it easy to form p-n junctions, thus suitable for use in a laboratory condition when designing new strategies of devices. However, as we commented in Q3 of Reviewer #2, the susceptibility of organic materials to high temperature and organic solvent critically limit the patterning by conventional photolithography. Consequently, the device scale is currently limited to the tens of μm level, determined just by the resolution of shadow mask patterning. However, this does not mean that the proposed device scheme is not scalable since other channel materials, which can produce NTC characteristics by forming reliable heterojunction, can be fully utilized for the semiconducting layer. Moreover,

the device design of the binary-ternary logic reconfigurable inverter is basically identical to that of the conventional CMOS-type inverter except that one transistor is replaced with the proposed heterojunction non-volatile memory transistor. Therefore, if channel materials compatible with the conventional photolithography are adapted, we expect that the state-of-the-art CMOS manufacturing process can be applied usually to fabricate the proposed device on a nanometer scale with the conventional integrated circuit design.

We revised the manuscript by adding the above discussion about the scalability of the proposed device as follows;

(1) Revised the main text in the revised manuscript (p. 3)

→ Notwithstanding the successful advancements in various applications such as logic circuits, physical/chemical sensors, memory and artificial synapse, low resistance of organic materials to high temperatures and solvents makes OTFTs incompatible with conventional photo-lithography process, limiting the integration density and information processing capability of organic ICs¹².

(2) Revised the main text in the revised manuscript (p.18)

→ Compared to conventional Si-CMOS technology, the proto-type logic devices demonstrated in this work shows relatively higher voltage, lower current, and larger size ... by using high- k inorganic dielectric materials, lithography-compatible semiconductors, or employing other device optimization strategies such as molecular/morphology engineering⁶⁵⁻⁶⁸, threshold voltage control^{69,70}, and contact resistance engineering^{69,71}.

(3) Added the reference [12] and [65-71] in the revised manuscript.

Q6) For real life applications, it would be of utmost importance to investigate the thermal stability of the device. Can the authors provide data at elevated temperatures? How does the PVR evolve with temperature?

Response:

Thank you very much for giving us the constructive suggestion. To investigate the thermal stability, we have measured the negative transconductance (NTC) characteristics of the heterojunction non-volatile memory transistor (H-MTR) by increasing the temperature in increments of 20 K (**Fig. R7**). Both the peak current (I_{Peak}) and valley current (I_{Valley}) were gradually increased as the temperature increased up to 358 K due to thermally activated transport in organic semiconductors^[R45]. After 378 K, the threshold voltage (V_{TH}) of the device was gradually increased and resulted in the decrease in I_{Peak} , which is attributed to thermal degradation of the p-type semiconductor, DNNT^[R46]. At 478 K, the device no longer exhibited switching characteristics. Although the peak-to-valley current ratio (PVCR) continuously decreased as the temperature increased, NTC characteristics could still be observed even at the temperatures up to $T = 458$ K, which demonstrates the potential of the proposed device concept for practical logic device applications.

Figure R7. Temperature-dependent NTC characteristics of H-MTR. a, $I_D - V_G$ plots and **b,** Extracted I_{Peak} and I_{Valley} and **c,** Extracted PVCR of H-MTR as functions of temperature.

We revised the manuscript by adding the above discussion about the thermal stability of the H-MTR as follows

(1) Revised the main text in the revised manuscript (p.13)

→ In terms of thermal stability, the NTC characteristics of the H-MTR were monitored by increasing the device temperature in increments of 20 K. Initially, ... NTC

characteristics could still be observed even for temperatures up to $T = 458$ K, which demonstrates the potential of the proposed device concept for logic device applications.

(2) Added the temperature-dependent NTC characteristics of the H-MTR (**Fig. R7**) to Supplementary Figure 7 in Supplementary Information (p. 11)

(3) Added the reference [63-64] in the revised manuscript.

[R45] *Nat. Commun.*, 2021, 12(1), 21.

[R46] *Nat. Commun.*, 2012, 3(1), 723.

Reviewer (#3)'s COMMENTS:

Authors reported a heterojunction non-volatile memory transistor (H-MTR) containing a drain-aligned floating gate to control the negative transconductance (NTC) characteristics. Finally, authors demonstrated ternary/binary dynamic logic conversion-in-memory to generate three different output logic sequences for the same input signal in three logic levels. My comments are follows.

Response:

We appreciate the Reviewer's encouraging comments. We present the point-by-point response for each comment as follows.

Q1) A drain-aligned floating gate is known and authors claims "newly developed". Please clarify

Response:

We agree with the Reviewer #3's comment about what we over-claimed. As pointed out by the Reviewer #3, terms such as "semi-floating gate" (*Nat. Nanotechnol.*, 2018, 13(5), 404-410) or "partial floating gate" (*Nat. Electron.*, 2022, 5, 752-760) have been used in previous reports, which can refer to "drain-aligned floating gate". The previous study using the termed "semi-floating gate" was about the high-speed and low-power quasi-non-volatile memory^[R47]. The device had a p-n junction switch at the source region to achieve fast charge transport, while the device had a flash memory structure at the drain region for charge storage. However, in the device, the floating gate was patterned under the whole channel as in conventional flash memory. The other previous study using the term "partial floating gate" was about reconfigurable transistor and reconfigurable non-volatile memory that showed either p-type or n-type operation, depending on the polarity of the trapped charges in the floating gate^[R48]. In the device, the floating gate was patterned not only under the drain but also under the source to achieve p-type/n-type reconfigurable operation.

In this work, floating gate was utilized in the context of asymmetric structure of a heterojunction transistor where p-type layer connects the source to drain electrode while the n-type layer interacts with only the drain electrode through the interposed p-type layer. By

designing the floating gate to be located only under drain electrode of heterojunction transistor, the amount of electron injection from the drain electrode to p-n heterojunction layer can be adjusted effectively, allowing the implementation of a heterojunction non-volatile memory transistor (H-MTR) with dynamic NTC characteristics. The dynamic NTC characteristics based on asymmetric floating gate have not been reported previously, and the function of the floating gate is completely different from that of the previous studies. So, we introduced the term "drain-aligned floating gate" to more accurately describe the proposed device and differentiate it from conventional "semi-floating gate" or "partial floating gate", even though the drain-aligned floating gate and previously reported "semi-floating gate" or "partial floating gate" share structural similarity. Considering the discussion above, we revise the sentence to clarify that the proposed device itself is newly developed, rather than solely the drain-aligned floating gate.

We revised the manuscript as follows;

(1) Revised the main text in the revised manuscript (p.1)

→ A new type of heterojunction non-volatile memory transistor (H-MTR) has been developed, in which the negative transconductance (NTC) characteristics can be controlled systematically by a drain-aligned floating gate.

(2) Revised the main text in the revised manuscript (p.19)

→ A drain-aligned floating gate heterojunction non-volatile memory transistor was proposed, where the FG is defined only under the drain electrode.

[R47] *Nat. Nanotechnol.*, 2018, 13(5), 404-410.

[R48] *Nat. Electron.*, 2022, 5, 752-760

Q2) The asymmetric device configuration of H-TR is expected to effectively adjust the amounts of electrons injected into the channel, and thus systematic control of NTC characteristics with programming operation can be achieved. Can authors comment on the p-type and n-type parameters at the drain side? How it affects the NTC characteristics?

Response:

Thank you very much for the Reviewer #3's constructive comment. To clarify the parameter characteristics of the p-type and n-type, we can describe the proposed H-MTR in an equivalent circuit as follows. In the H-MTR device, there are two charge transport paths: one through a lateral p-n junction (Path 1) and the other through a continuous p-type layer (Path 2), as shown in **Fig. R8a**. The electrical characteristics of the top p-type layer (P2) that forms a vertical heterojunction with the n-type layer (N1) are different from those of the bottom p-type layer (P1) in contact with the dielectric layer. Therefore, the equivalent circuit model of the H-MTR consists of cascaded P1 and N1 transistors and cascaded P1 and P2 transistors. The serially connected transistors (P1-N1 and P1-P2) are also connected in parallel (**Fig. R8b**). The ideal transfer characteristics of the H-MTR can be represented by the equivalent circuit as shown in **Fig. R8c**. It has three nominal threshold voltages: the threshold voltage of P1 transistor (V_{TH_P1}), the threshold voltage of P2 transistor (V_{TH_P2}), and the threshold voltage of N1 transistor (V_{TH_N1}). From the transfer characteristics, V_{TH_P1} and V_{TH_P2} determine peak voltage (V_{Peak}) and peak current (I_{Peak}) while V_{TH_P1} and V_{TH_P2} govern valley voltage (V_{Valley}) and valley current (I_{Valley}) of the H-MTR. Since drain-aligned floating gate can adjust the V_{TH_P2} and V_{TH_N1} of the H-MTR according to its memory state, the NTC characteristics of V_{Peak} , V_{Valley} , I_{Peak} , I_{Valley} can be effectively controlled by programming operation. Additionally, N2 transistor is responsible for drain current in NTC region. Therefore, the magnitude of NTC is determined by transconductance (g_m) of the N1 transistor, which in turn, is affected by charge carrier mobility of the N1 transistor.

Figure R8. **a**, Schematic diagram illustrating charge transport paths, **b**, equivalent circuit model, and **c**, corresponding transfer characteristics of the H-MTR.

We revised the manuscript by adding the above discussion on the relationship between NTC characteristics and each parameter of the p-type and n-type at the drain side of the proposed device as follows;

(1) Revised the main text in the revised manuscript (p.10)

→ The former one is originated from the charge carrier transport through the lateral DNNT/PTCDI-C13 junction near the gate dielectric and corresponded to the V_{TH} before the NTC region (namely, V_{TH_P1}) while the latter stems from the charge carrier transport through the DNNT “back channel”, corresponding to the V_{TH} exceeding the NTC region (namely, V_{TH_P2}) (Supplementary Fig. 4).

(2) Added a schematic diagram illustrating equivalent circuit model (Fig. R8b), and

corresponding transfer characteristics (**Fig. R8c**) of the H-MTR to Supplementary Figure 4 in Supplementary Information (p. 8).

(3) More detailed discussion on the relationship between NTC characteristics and each parameter of the p-type and n-type at the drain side was added in Supplementary Information (p. 8)

→ In H-MTR, there are two different charge carrier transport paths. Each path corresponds to charge flow through p-type and n-type lateral junction (Path 1) ... H-MTR according to its memory state, the NTC characteristics of V_{Peak} , V_{Valley} , I_{Peak} , I_{Valley} can be effectively controlled by programming operation.

Q3) For non-technical readers, it would be good to have a schematic diagram to explain the concept from device physics to achieving reconfigurable binary/ternary logic conversion.

Response:

As the Reviewer #3 constructively suggested, we added a conceptual schematic that demonstrates how binary/ternary logic conversion can be implemented based on the device physics and transfer characteristics of H-MTR (**Fig. R9**). The device physics is reflected on **Fig. R9a** and **Fig. R9b**, where electron injection can be facilitated or hindered by drain-aligned floating gate-mediated gate-to-drain electric field, which in turn increases or decreases the range of NTC region, resulting in dynamic NTC characteristics of H-MTR. The implementation of reconfigurable binary/ternary logic conversion is demonstrated in relation to the device physics and dynamic NTC characteristics of H-MTR (**Fig. R9c**), where an intermediate logic state of the binary/ternary reconfigurable logic inverter is determined depending on the NTC characteristics of the H-MTR.

Additionally, we introduced schematic illustration as well as energy band diagram describing the charge carrier transports to supplement the explanation of device physics and help reader's understanding (**Fig. R10**) at the specific gate voltages ($v_i \sim v_i$) (**Fig. R10c**) of the measured transfer characteristics for (+) programmed or (-) programmed H-MTR (**Fig. R10a, b**).

We revised the manuscript as follows;

(1) Revised the main text in the revised manuscript (p.10)

→ Based on the capability to generate or eliminate the NTC characteristics of the H-MTR, the R-inverter can be implemented by connecting ... show the comparable conductance is limited to a few points, resulting in binary logic inverter without an intermediate logic state. (Fig. 1e, right).

(2) Revised the main text in the revised manuscript (p.12)

→ A Schematic illustration and energy band diagram describing the charge carrier transport in H-MTR according to its memory state are provided in Supplementary Fig. 5.

(3) Added the conceptual schematic illustrating the operating principle of the binary/ternary reconfigurable logic inverter (**Fig. R9**) to Figure 1 of the manuscript. (p. 28)

(4) Added the schematic illustration and energy band diagram describing the charge carrier transports in H-MTR (**Fig. R10**) to Supplementary Figure 5 in Supplementary Information (p. 9)

Figure R9. A schematic diagram illustrating the concept of device physics and binary/ternary logic conversion. a, A schematic illustrating the charge injection at drain electrode and resultant transfer characteristic of H-MTR for (+) programmed state and **b**, for (-) programmed state. **c**, A schematic diagram illustrating the operating principle of the R-inverter.

Figure R10. Schematic illustration and energy band diagram describing the charge carrier transports in H-MTR according to its memory state. a, The transfer characteristics of H-MTR with respect to programming voltage (V_{prg}). $V_{prg} = -15$ V and **b,** $V_{prg} = +18$ V. A dashed curve represents transfer characteristics of H-MTR in pristine state. **c,** Schematic illustrations and the energy-level alignments of pristine, (+) programmed, and (-) programmed H-MTR at specific gate voltage (V_G).

Q4) For +ve programming operation, the drain current is confined in one line. Can authors explain this phenomenon?

Response:

Thank you very much for giving us the valuable comment. To clarify the mentioned phenomenon, we introduced schematic illustration and energy band diagram that describe the charge transport in the H-MTR for (+) programming operation (**Fig. R11**). As we mentioned in the answer for Q2 of Reviewer #3, there are two different charge transport paths in H-MTR; charge flow through p-type and n-type lateral junction (Path 1), and charge flow through continuous p-type layer (Path 2) (**Fig. R8a**). When the drain-aligned floating gate is negatively charged under (+) programming operation, electron injection from the drain electrode to n-type semiconductor is limited by the decreased gate-to-drain electric field (E_{GD}). This leads to electron depletion of the n-type semiconductor, and thus charge carriers cannot transport through p-type and n-type lateral junction (Path 1) (**Fig. R11d**). Meanwhile, charge carriers can still be transported through the continuous p-type layer (Path 2) (**Fig. R11e**). This Path 2 involves two different p-type semiconductors, bottom p-type semiconductor (P1) in touch with dielectric layer and top p-type semiconductor (P2) forming vertical heterojunction with underlying n-type semiconductor (N1), thereby, generating energy offset required for charge transport from P1 to P2 at the above of lateral junction edge of the p-type and n-type semiconductor. This energy offset is reflected in V_{TH_P2} (**Fig. R11a**) and cannot be lowered by (+) programming operation, as the drain-aligned floating gate cannot affect the lateral junction edge of the p-type and n-type semiconductor. Consequently, charge transport through Path 1 is suppressed, while charge transport through Path 2 remains unchanged, resulting in monolithically current-increasing transfer characteristics with the threshold voltage of V_{TH_P2} (**Fig. R11b**).

We revised the manuscript by adding the above discussion about the (+) programming operation of H-MTR as follows;

(1) Revised the main text in the revised manuscript (p.12)

→ A Schematic illustration and energy band diagram describing the charge carrier

transport in H-MTR according to its memory state are provided in Supplementary Fig. 5.

(2) Added the schematic illustration and energy band diagram describing the charge carrier transports in H-MTR (Fig. R10) to Supplementary Figure 5 in Supplementary Information (p. 9)

Figure R11. Schematic illustration and energy band diagram describing the charge transport in H-MTR for (+) programming operation. a, Schematic transfer characteristics of the H-MTR before programming operation and **b**, after (+) programming operation. **c**, Schematic illustration and the energy-level alignment of H-MTR at $V_G < V_{TH_P1}$, **d**, $V_{TH_P2} < V_G < V_{TH_P1}$, and **e**, $V_G > V_{TH_P2}$.

REVIEWERS' COMMENTS

Reviewer #1 (Remarks to the Author):

The authors have addressed all the points raised by the reviewer in a satisfactory manner. The reviewer can now recommend publication.

Reviewer #2 (Remarks to the Author):

The authors have answered all questions raised by the reviewers. I would suggest to publish the manuscript.

Response to the Reviewers' Comments

The authors thank the Reviewers for their considerate review of our manuscript and the valuable comments.

Reviewer (#1)'s COMMENTS:

The authors have addressed all the points raised by the reviewer in a satisfactory manner. The reviewer can now recommend publication.

Response:

We appreciate the Reviewer for the valuable comments.

Reviewer (#2)'s COMMENTS:

The authors have answered all questions raised by the reviewers. I would suggest to publish the manuscript.

Response:

We appreciate the Reviewer for the valuable comments.